# Diagnostics of Sacroiliac Joint Differentials to Axial Spondyloarthritis Changes by Magnetic Resonance Imaging

**DOI:** 10.3390/jcm12031039

**Published:** 2023-01-29

**Authors:** Anne Grethe Jurik

**Affiliations:** 1Department of Radiology, Aarhus University Hospital, Palle Juul-Jensens Boulevard 99, 8200 Aarhus, Denmark; anne.jurik@aarhus.rm.dk; 2Department of Clinical Medicine, Aarhus University, Palle Juul-Jensens Boulevard 82, 8200 Aarhus, Denmark

**Keywords:** imaging diagnostics, axial spondyloarthritis, magnetic resonance imaging, diagnosis, differential

## Abstract

The diagnosis of axial spondyloarthritis (axSpA) is usually based on a pattern of imaging and clinical findings due to the lack of diagnostic criteria. The increasing use of magnetic resonance imaging (MRI) of the sacroiliac joints (SIJ) to establish the diagnosis early in the pre-radiographic phase has resulted in a shift in the paradigm with an increasing frequency of axSpA diagnoses and a changed sex distribution. Non-radiographic axSpA affects males and females nearly equally, whereas ankylosing spondylitis predominantly occurs in males. The MRI-based increasing frequency of axSpA in women is mainly due to the presence of subchondral bone marrow edema (BME) on fluid-sensitive MR sequences, which may be a non-specific finding in both women and men. Due to the somewhat different pelvic tilt and SIJ anatomy, women are more prone than men to develop strain-related MRI changes and may have pregnancy-related changes. Awareness of non-specific subchondral BME at the SIJ is important as it can imply a risk for an incorrect SpA diagnosis, especially as the clinical manifestations of axSpA may also be non-specific. Knowledge of relevant MRI and clinical features of differential diagnoses is needed in the diagnostic workout of patients with suspected axSpA considering that non-SpA-related SIJ conditions are more common in patients with low back or buttock pain than axSpA sacroiliitis. The purpose of this review was to present current knowledge of the most frequent differential diagnoses to axSpA sacroiliitis by MRI taking the clinical characteristics into account.

## 1. Introduction

Inflammatory changes at the sacroiliac joint (SIJ) are a hallmark of axial spondyloarthritis (axSpA), which is usually divided into radiographic axSpA and non-radiographic axSpA depending on radiographic changes in accordance with the modified New York criteria for ankylosing spondylitis [1,2]. Detection of sacroiliitis before the occurrence of radiographic SIJ changes demands visualization by magnetic resonance imaging (MRI) of the preceding inflammatory changes. MRI findings supporting active inflammatory axSpA changes primarily consist of subchondral bone marrow edema (BME) on fluid-sensitive MR sequences, such as STIR (short tau inversion recovery) and T2-weighted fat-suppressed (FS) sequences. In addition, MRI can visualize other signs indicating active inflammation, such as inflammation at the site of erosion, fluid or enhancement in the joint space, capsulitis and enthesitis, and also structural axSpA changes such as erosions, subchondral fat metaplasia, fat metaplasia in an erosion cavity (backfill), sclerosis and joint space alterations, including widening, narrowing, bone bud and ankylosis [3] (Figure 1 and Figure 2).

During the last 20–30 years, MRI has therefore been increasingly used to diagnose axSpA and is an essential part of the 2009 ASAS (Assessment of SpondyloArthritis international Society) axSpA classification criteria [4] with a subsequent update of the MRI criteria for active sacroiliitis in 2016, which attracted attention to the importance of concomitant structural changes to support the presence of active inflammation [5]. Use of MRI in the diagnosis of axSpA has resulted in a shift in the paradigm with an increasing frequency of axSpA diagnoses due to the addition of patients with non-radiographic axSpA sacroiliitis changes by MRI and normal or equivocal pelvic radiographs. Moreover, the sex distribution has changed. Ankylosing spondylitis, diagnosed on the basis of structural SIJ changes by radiography, is most frequent in males; non-radiographic axSpA affects males and females nearly equally [2,6].

However, signs compatible with sacroiliitis are not specific for axSpA occurring in other conditions and even in healthy individuals [7,8,9,10,11,12,13,14,15,16]. This implies a risk for an incorrect SpA diagnosis as the clinical manifestations of axSpA may also be non-specific. Knowledge of relevant differential diagnoses and their imaging features is therefore important in the diagnostics of patients with suspected axSpA, considering that non-SpA-related SIJ conditions are more common in patients with low back or buttock pain than axSpA sacroiliitis [17,18]. MRI findings at the SIJ in patients with low back pain and suspected sacroiliitis were assessed in two cohorts: one where the patients were referred to MRI by doctors from different specialties, such as rheumatologists, orthopedic surgeons or family physicians [17], and one encompassing only patients referred by a rheumatologist with inflammatory back pain beginning before the age of 40 years [18]. A summary of the results appears in Table 1. The prevalence of axSpA was highest in the cohort of patients referred by rheumatologists, 36% and 25%, respectively. A predominance of normal findings and other disorders emphasized that clinical findings and symptoms such as inflammatory back pain in itself are not specific even when assessment was made by experienced rheumatologists. This, in addition to the possibility of non-specific MRI findings, imply a need for a close collaboration between rheumatologists and radiologists to establish a correct diagnosis of axSpA or its differentials taking all clinical and imaging aspects into account [19].

The purpose of this review was to present the current knowledge of the most frequent differential diagnoses to axSpA sacroiliitis by MRI as well as their clinical characteristics, including a description and illustration of typical MRI appearances of changes simulating axSpA sacroiliitis. Primarily, this review includes imaging of adults, but includes a brief description of growth-related features in childhood using the following headings:

## 2. SIJ Appearance in Childhood

Increased signal intensity in the subchondral bone marrow on fluid-sensitive MR sequences compared to the inter-foraminal sacral bone marrow is normal in children before skeletal maturity, typically occurring at the age of 15 years in girls and 17 years in boys. Both the sacrum and the iliac bones develop by endochondral bone formation with the conversion of cartilage to bone occurring in the subchondral areas of the SIJ. The zone of primary spongiosa with ongoing calcification is, as other developing endochondral bone areas, vascularized and thus shows increased signal intensity on fluid-sensitive sequences (STIR/T2FS) equivalent to the high signal intensity in metaphyseal growth areas. In children, it is therefore rational to compare the marrow signal intensity adjacent to the sacroiliac joint space with other growth areas, most obviously the metaphyseal equivalent area at the posterior iliac apophyses [20] instead of the inter-foraminal reference used in adults [4,5]. However, the iliac crests are some of the last apophyseal areas to close and the subchondral signal intensity at the SIJ is therefore normally less or equal to the signal intensity at the posterior iliac crest apophyses [21] (Figure 3). Important in the differentiation from inflammatory BME changes, the increased signal intensity at the SIJ on STIR/T2FS is characteristically symmetric with a sacral predominance whereas asymmetrical BME or BME predominance in the ilium raises suspicion of inflammatory BME [22].

The development of the SIJ includes a gradual fusion of sacral vertebral segments as well as the sacral apophyses. Fusion of the sacral segments starts between S1 and S2 followed by the intervertebral space S2/S3, which can remain incomplete up to 18 years in both genders [23]. There are multiple small apophyses in the joint space, which ossify slowly throughout adolescence and fuse with the sacrum, a process finishing earlier in girls than in boys [22,23,24]. Intraarticular nuclei in childhood have been observed up to the age of 18 years [23] and may occasionally persist in adulthood [25,26]. This growth process contributes to a frequently observed high signal in the joint space and/or in the subchondral bone on T2/STIR and postcontrast images [27] (Figure 4), whereas focal, deep and intense edema and/or contrast enhancement at the SIJ, features of juvenile sacroiliitis, do not occur in healthy children [27]. The growth processes also imply a frequent occurrence of irregular articular SIJ margins with lack of well-defined cortical bone on T1-weighted and cartilage sequences, which can mimic erosions [28].

## 3. Normal SIJ Anatomy in Adults and Technical MRI Aspects

Knowledge of the normal anatomy and appearance of the SIJ by MRI is important in the interpretation of images. The SIJs are formed between the sacrum and the iliac bones and built to transfer all forces from the spine to the lower extremities. The sacrum is wider superiorly than inferiorly and also wider anteriorly than posteriorly, and joint stability is secured by the sacrum being wedged cranially and dorsally between the iliac bones within the pelvic ring accompanied by strong surrounding ligaments [29]. The SIJ consists of two compartments, a ventral cartilaginous and a dorsal ligamentous compartment [30]. The cartilaginous joint surfaces vary considerably in shape and contour, especially in women, and may change with age, but are without major protrusions. Due to a more backward tilted sacrum, the cartilaginous joint surface is more horizontally located, and it is usually smaller in women than in men [29] (Figure 5). The SIJ in males is typically formed corresponding to the sacral segments S1, S2, and S3, whereas inclusion of the whole S3 segment is uncommon in females [29]. The corresponding iliac joint facets have the opposite contour resulting in a slim joint space with a width of 2 to 3 mm in young adults and gradually narrowing with age [31,32]. The cartilaginous joint compartment can be classified as a symphysis with a few characteristics of a synovial joint confined to the distal portion of the joint, whereas the ligamentous compartment contains strong ligaments contributing to joint stability [30].

Adequate visualization of both joint compartments demands sequences in two perpendicular planes, semi-coronal and semi-axial planes, respectively (Figure 6 and Figure 7), which is a prerequisite for the precise location of joint abnormalities [30,33]. Thus, the current minimum standard for a diagnostic SIJ MRI includes at least four sequences: a semi-coronal T1-weighted sequence sensitive for marrow fat signal and a sequence designed for optimal visualization of the bone-cartilage interface (articular surface), such as a T1 fat-saturated (T1FS) sequence or a gradient echo sequence (depending on MR-equipment), and fluid-sensitive sequences such as STIR or T2FS in two perpendicular planes (semi-coronal and semi-axial) to detect BME [34,35,36] (Figure 6). Optimally, the semi-axial field of view should be centered to allow additional visualization of the symphysis and the upper part of the hips to screen for concomitant changes in these areas, which can be affected in axSpA and thereby also contribute to the detection of differential diagnoses (Figure 2 and Figure 7). More advanced MR-sequences, such as DWI (diffusion-weighted imaging,) have been studied without confirming certain diagnostic advantages compared to the sequences mentioned above.

In the interpretation of the different sequences, it is important to be aware of the location of the different joint portions on the semi-coronal and semi-axial slices, respectively. The most anterior coronal slice in the SIJ will typically be in middle portion of the cartilaginous joint compartment, especially in women, whereas the upper and lower joint portions are located more posteriorly and usually appear on slices also showing areas of the ligamentous joint compartment, as shown in Figure 7.

MR signal intensity in the bone marrow varies in healthy persons both on STIR/T2FS and T1-weighted sequences [16,37]. Minor areas of increased subchondral signal on STIR/T2FS often occur in individuals above 30 years [16]. In a study of 94 healthy individuals aged 20–49 years, BME was detected in all age groups with an increasing prevalence with age from 13.9% in subjects aged 20–29 years to 35.7% in subjects aged ≥ 40 years. However, the extent was limited, and only one subject in the age group 20–29 had a positive MRI for active sacroiliitis according to the ASAS definition [5]. A total of 17.2% of the subjects ≥ 30 years old fulfilled the definition of a positive MRI for sacroiliitis, but deep and/or intense BME was relatively rare. The SIJ BME detected based on the semi-coronal slices most frequently occurred in the superior portion of the sacrum (both anterior and posterior) followed by the inferior ilium. A comparable prevalence was reported in a study by de Winter et al. [10], including 47 healthy individuals aged 18–45 years with a mean age of 31 years. In this study, 23.4% had BME fulfilling the ASAS criteria for sacroiliitis, most frequently located to the inferior ileum; deep and intense BME did not occur. The location of BME to the upper sacrum and/or inferior ileum in these studies based on semi-coronal STIR images indicates that the BME is predominantly caused by strain [15], but occasionally the physiologic subchondral BME occurring in children/adolescents can persist in early adulthood and may be a pitfall in the detection of sacroiliitis [37].

On T1-weighted images there can be a patchy distribution of fat within the bone marrow with an increasing prevalence with increasing age, but not manifesting in subchondral areas with fat metaplasia as seen in ankylosing spondylitis [16,37,38]. The reported prevalence of fat deposition varies. In a study of 94 healthy subjects aged 20–49 years [16], fat metaplasia was detected across all age groups in 13.7%, but none of the heathy subjects aged 20–29 years had fat metaplasia in three or more of the scoring areas (≥3 quadrants [39]). However, in a study of 485 non-rheumatological patients [38], fat metaplasia was common with a prevalence of 50.6% in the age groups < 45 years increasing to 94.4% in patients ≥ 75 years. The observed predominantly focal or patchy fat depositions can be considered non-specific findings, especially seen in a degenerative setting and in older healthy individuals as confirmed in other studies [37]

Erosion or erosion-like lesions can also be a normal finding, but the observed prevalence has varied considerably, probably due to different definitions. Thus, in the study of 94 healthy subjects aged 20–49 years [16], SIJ erosions were detected across all age groups in 20%, but none of the heathy subjects aged 20–29 year fulfilled the proposed structural criteria for axSpA sacroiliitis using cut-off values for erosions and fat metaplasia with a specificity ≥ 95% for axSpA (erosions in ≥3 quadrants, fat metaplasia in ≥3 quadrants, or erosions and/or fat metaplasia in ≥5 quadrants [39]). The prevalence of erosions also increased with age, and erosions were observed in 39.3% of individuals aged ≥ 40 years, and 17.9% of them had erosions in ≥3 quadrants [16]. In contrast, erosions were uncommon in the study group of 485 non-rheumatological patients, only occurring in 0.6% of patients < 45 years of age and in 2.6% of the entire study population [38].

Data are scarce on the presence of sclerosis, osteophytes, and joint space alterations at normal SIJ by MRI, but sclerosis and osteophytes were reported in 13.7% and 37.0% of the 485 non-rheumatological patients, respectively [38]. These features have mainly been studied by CT where osteophytes were detected even in young patients with an increasing prevalence with advancing age [31]. Subchondral iliac sclerosis > 5 mm and sacral sclerosis > 3 mm was rare below the age of 40 years but occurred with increasing frequency in persons aged above 40 years, most frequently in obese and multiparous women [31]. The normal joint space width assessed by CT was observed to decrease with increasing age, with mean values being 2.3 ± 0.4 and 1.9 ± 0.2 mm before and after 40 years of age, respectively [31]. The joint width is difficult to measure on T1-weighted sequences and a cartilage sequence is needed; however, it may still be difficult to evaluate because the joint space varies through the joint.

The imaging features of non-specific BME, fat deposition, and sclerosis appear in the figures in Section 3 and Section 4.

## 4. Anatomical Variations

### 4.1. Well Described SIJ Variants

Anatomical SIJ variations are frequent findings in adults, especially women [40] and awareness of their presence is important when interpreting SIJ MRI as they may represent pitfalls. Seven SIJ variations have been described: accessory SIJ, iliosacral complex, bipartite iliac bony plate, semicircular defects, crescent-like iliac bony plate, dysmorphic SIJ, and unfused nuclei at the sacral wing (Figure 8, Figure 9 and Figure 10) [40]. The detection and description of SIJ variations have primarily been based on CT [25,26,41,42,43,44], but they are detectable by MRI [40,45], which can additionally visualize concomitant BME, sclerosis, and/or fat deposition. Only two of these variants, dysmorphic SIJ and unfused nuclei, occur at the cartilaginous SIJ compartment and may therefore directly cause MRI changes simulating sacroiliitis confined to the cartilaginous joint compartment [39]. Dysmorphic SIJ changes characterized by altered joint form are directly visible (Figure 8F and Figure 9), but there may be concomitant irregular joint facets simulating sacroiliitis changes necessitating supplementary CT to confirm the absence of erosive changes (Figure 9). Persistence of unfused nuclei in the joint space may also present pitfalls and is therefore described and illustrated separately in detail. BME adjacent to accessory SIJ and iliosacral complex will be located to the ligamentous joint compartment where BME areas will not be mistaken for sacroiliitis changes. However, the presence of variants in the ligamentous compartment may elicit strain-related changes in the cartilaginous joint compartment, often located anteriorly in the sacrum [40]. BME in this location should thus be interpreted cautiously in patients presenting with low back pain.

### 4.2. Persistence of Unfused Nuclei in Adulthood

Nuclei at the upper sacral corners (sacral wing) have been reported to be the last to fuse with the sacrum and may persist in adulthood (Figure 10), usually without any symptoms.

Persistence of unfused nuclei in other parts of the joint space may occur, usually at the border between S1 and S2 where atypical osseous fusion may also occur (Figure 11, Figure 12 and Figure 13). There can be concomitant BME and also erosion-like lesions mimicking sacroiliitis, which together with accompanying low back and/or buttock pain can make the differentiation from axSpA sacroiliitis difficult (Figure 11, Figure 12 and Figure 13).

Knowledge about the clinical importance of SIJ variations is limited and should be established, including relation to imaging findings. Only accessory SIJs seem to have been investigated to some extent, probably because they are frequent, reported present in 13–19% of the general population, and occurring bilaterally in 50% of the affected persons [25,26,29,44]. The prevalence of accessory SIJ has been reported to increase with age and be highest in obese individuals above the age of 60 years [26], indicating that strain on the joint may play a role. Moreover, secondary degenerative changes directly related to the accessory SIJ have been described based on imaging findings as well as specimens, consisting of subchondral BME, sclerosis, cysts, and osteophytes [47,48,49,50]. However, it must be determined whether complaints of chronic or recurrent SIJ pain in the presence of accessory SIJ with surrounding active inflammation and/or structural changes by MRI correlates with the clinical findings. The persistence of unfused nuclei in the joint space may be related to pain (Figure 11, Figure 12 and Figure 13), but occurrence and associated clinical findings have to be further studied, including a detailed delineation of the joint space using axial MRI slices and preferably supplementary low dose CT examinations.

### 4.3. Lumbosacral Transitional Vertebrae

Lumbosacral transitional vertebra is a common variation, either encompassing the sacralization of the fifth lumbar vertebral body or lumbarization of the S1 segment, with a wide variety of morphologies ranging from broadened transverse processes to complete fusion occurring bilaterally, symmetrically, or asymmetrically as well as unilaterally [51]. Transitional vertebrae occur in up to 25% of the population [52] and may be accompanied by low back pain, especially in the case of vertebrae with bilateral persistent cleavage between the transverse process and the upper border of the sacrum (Castellvi type I and type II [51]). Concomitant BME, fat deposition, and/or sclerosis may occur, but is usually confined to pseudo-articulations and as such only occasionally reaches the SIJ possibly causing confusion with axSpA sacroiliitis changes [52] (Figure 14).

## 5. Osteitis Condensans ilii and Pregnancy-Related Changes

### 5.1. Osteitis Condensans ilii (OCI)

OCI is originally defined based on its radiographic appearance with unilateral or bilateral triangular-shaped sclerosis in the ileum corresponding to the weight-bearing portion of the SIJ, spared SI joint space, and no evidence of erosions [53]. The estimated prevalence of OCI in the general population is between 0.9% and 2.5% [53,54,55]. OCI can be an asymptomatic incidental finding but has been observed in 2.5–8.9% of patients presenting with low back or SIJ pain [17,18,56]. It occurs predominantly in females, especially postpartum women [55], but is occasionally diagnosed in men and nulliparous women [57] where it is often related to obesity or excessive physical load to the SIJ, e.g., due to scoliosis. Characteristically there are normal inflammatory biomarkers and HLA B27 negativity [57]. The pathophysiology of OCI is not fully elucidated, but previous pregnancy-related strain to the SIJ often seems to play a role, probably combined with hormonal changes. The increased tilt of the pelvis in women compared to men and the frequent occurrence of anatomical variations in females [40] may also be involved.

The MRI appearance of OCI is characterized by manifest subchondral iliac sclerosis displaying low signal intensity on all sequences, usually located anteriorly with a mean thickness of approx. 13 mm according to the analysis of Ma et al. [57]. There is often concomitant BME, reported present in 48–93% of cases [57,58]. The iliac BME in OCI usually has a characteristic continuous distribution and is located peripherally to the subchondral sclerosis [57] (Figure 15) whereas BME as part of axSpA changes is usually discontinuous, extending to other parts of the joint outside the load-bearing areas [57]. However, sacral BME is also frequent in patients with OCI, often located to the strain-related areas anteriorly (Figure 15). The BME as part of OCI may both display high signal intensity on STIR/T2FS and have a depth above 10 mm [57], which are features often seen in active axSpA sacroiliitis. In addition, OCI changes may be accompanied by low back pain and can therefore also clinically be difficult to distinguish from axSpA. Only the manifest sclerosis, absence of definite erosion, and especially the anterior location of BME changes are valid in the differentiation from sacroiliitis. In a recent study of 27 patients with OCI (96.7% women) and 27 axSpA patients (46.7% men) matched by back pain period, BME occurred in 92.6% of patients with OCI and 85.5% of patients with axSpA [58]. Nearly all BME lesions in OCI were located at the anterior part of the SIJ, whereas axSpA lesions predominantly occurred in the middle part of the joint [58]. Moreover, erosions were observed in only 7.4% of patients with OCI compared with 66.7% in axSpA. Clinically there were also notable differences with a lower prevalence of inflammatory back pain and HLA-B27 in OCI compared with axSpA patients. However, the symptoms related to OCI can vary considerably from manifest low back pain to being asymptomatic. This may reflect a dynamic condition with periods of activity associated with SIJ pain and tenderness accompanied by SIJ BME shifting to a clinically inactive, asymptomatic stage with only structural imaging changes [54]. Such an undulating OCI course may clinically and by imaging sometimes mimic an inflammatory disorder and thus complicate the differentiation from axSpA changes. However, in general, OCI is the most probable diagnosis in cases of recent childbirth, when relatively preserved joints margins are evident, periarticular changes are confined to the anterior portion of the SIJ and no other symptoms or signs of SpA are present such as uveitis, inflammatory bowel disease, arthritis, dactylitis, or psoriasis.

### 5.2. Pregnancy-Related Changes

Pregnancy-related BME is frequent and has been reported in several studies, occurring both with and without pain [7,8,9,10,14]. The occurrence of BME seems to start already during pregnancy. In a longitudinal study including pregnant as well as postpartum women, SIJ BME was observed during pregnancy with an increased prevalence three months after childbirth followed by a gradual decrease. At 12-month postpartum, subchondral BME occurred in approx. 40% of the women sometimes accompanied by erosion-like lesions, sclerosis, and fat deposition [59]. The extent of BME may be as pronounced as in axSpA [8,59], occasionally with both intensity and depth > 1 cm [10,59]. This implies a risk of establishing an incorrect axSpA diagnosis in patients with low back pain at least 12 months postpartum and therefore constitutes an important differential diagnosis to axSpA changes. However, BME is usually detected in the strain-related area anteriorly, especially the middle anterior portion of the joint as shown in Figure 16.

Pregnancy-related changes can evolve into OCI with postpartum sclerosis and BME occurring anteriorly in the ileum, the location typical of OCI changes, as well as in the sacrum [7] (Figure 17). 

Consistent with the findings in OCI, erosion-like lesions are less frequent seen in pregnancy-related changes compared with axSpA, reported in 0–19% [8,9,14,59]. Backfill and ankylosis have been reported absent [8,14], but areas with BME may convert to areas with fat metaplasia [9,59].

The clinical importance of postpartum SIJ BME in non-SpA individuals has not been established although it occurs in symptomatic as well as asymptomatic postpartum women [8,14]. However, current knowledge of a high prevalence of BME even at 12 months postpartum underlines the importance of a history of previous pregnancies when evaluating SIJ MRI in postpartum females with pain to avoid an incorrect axSpA diagnosis [60].

## 6. Other Strain-Related SIJ Changes

Changes related to SIJ strain are most frequent in the load-related areas of the joint at the middle anterior joint portion [13] where subchondral BME, sclerosis, and/or fat metaplasia in addition to erosion-like irregularities may occur (Figure 18). Consistent with this, BME at the SIJ has been reported relatively frequently in patients with mechanical low back pain [12,13,19], and also in healthy, especially sports active persons [8,10,11,15]. The BME observed is usually minimal compared to BME in axSpA, but a study by de Winter et al. [10] reported changes to be highly suggestive of axSpA according to the ASAS criteria [5] in 12.5% (3/24) of runners and 6.4% (3/47) of patients with chronic back pain as well as in 23.4% (11/47) of healthy volunteers and in 57.1% (4/7) of women with postpartum back pain compared with 91.5% (43/47) of patients with axSpA [10]. However, deep BME lesions were not found in patients with chronic back pain, runners, or healthy volunteers, but were observed in 89.4% (42/47) of patients with axial SpA and in 1 of 7 women with postpartum back pain [10].

Based on semi-coronal MR slices, BME at the SIJ fulfilling the ASAS 2009 criteria was observed in 35% (9/20) of healthy hobby runners and in 41% (9/22) of ice-hockey players [15]. This non-inflammatory BME was most frequent in the posterior lower ilium followed by the anterior upper sacrum [15]. Concomitant use of semi-axial STIR images reduced the detection of subchondral BME in the lower posterior ileum because the axial slices visualized some of the BME changes to be due to partial volume effect of vascular signals, deep iliac ligament insertion containing vascular signals, or fluid-filled bone cysts giving the impression of SIJ BME on semi-coronal images [33]. The reduced frequency of confirmed BME in the lower ileum implied that the anterior sacrum became the most frequent site of strain-related BME, which based on the axial slices was located to the middle portion of the joint. Corresponding results based on semi-coronal images were obtained in a study of military recruits without back pain. Among these, 22.7% (5/22) and 36.4% (8/22) had BME fulfilling the ASAS MRI criteria before and 6 weeks after intensive physical training, respectively [11].

Data regarding strain-related structural changes have varied. In the study of hobby runners and ice-hockey players [15] erosions were rare with only one erosion detected in three runners and one ice-hockey player, respectively [15]. However, erosions were frequently detected in the military recruits, as 27.2% (16/22) had structural lesions both at baseline and follow-up, 13.6% had erosion-like lesions, 13.6% fat metaplasia, and 4.5% sclerosis, respectively.

MRI features, which may aid the differentiation between axSpA and strain-related SIJ BME, are concomitant entheseal changes occasionally seen in axSpA located to the ligamentous joint compartment and/or the iliac crest/wings, but are rare in other conditions [13,61]. Adequate visualization of these areas demands axial slices.

The findings in healthy individuals and in persons with strain-related SIJ changes underscore the importance of interpreting SIJ MRI findings in the appropriate clinical context, in young active individuals as well as in elderly. Thus, age, history, clinical features, the topographic location/distribution of BME, concomitant structural, or entheseal changes should be considered, being aware that non-inflammatory BME is most common in the anterior middle sacral portion of the joint.

## 7. Degenerative SIJ Changes/Osteoarthritis

Degenerative SIJ changes are rare below 30 years of age, but the possibility should be taken into account already in the fourth decade [62], and degenerative changes are common in middle-aged and older individuals both with and without accompanying symptoms [50,62]. The imaging findings have mainly been described based on CT features consisting of joint-space narrowing to less than 2 mm, subchondral sclerosis, which is usually dense, well-defined and with variable thickness, and osteophyte formation in addition to the occasional occurrence of intraarticular air (vacuum phenomenon), subchondral cysts, ankylosis, and sometimes small erosion-like lesions, but not manifest erosions as in axSpA (Figure 19).

Degenerative changes are especially common in patients with scoliosis (Figure 19) and developmental anomalies such as transitional vertebrae and accessory SI joints. In a recent study, the occurrence of anatomical joint variation and gender were observed to influence the development and location of degenerative SIJ changes. Ventral osteophytes occurred predominantly in men whereas dorsal osteophytes predominate in women, often located to areas of joint variants such as accessory joints and iliosacral complex [50]. The presence of accessory joints as well as bipartite iliac bone plate was frequently associated with sclerosis at the anterior part of the joint where strain-related changes are common [50].

Unfortunately, degenerative changes can sometimes be difficult to detect with certainty by MRI where the exact joint space width can be difficult to measure without appropriate cartilage sequences and small osteophytes may vanish due to the relatively thicker MR slices compared to CT slices. In addition to the changes best documented by CT, MRI may reveal dispersed fat deposition and/or subchondral BME usually occurring at the anterior portion of the SIJ adjacent to sclerotic areas, directly in the subchondral area and/or at areas with osteophytes. However, BME is usually minimal, often visible on only one or two slices. Thus, in middle-aged and elderly individuals with subchondral BME, fat deposition and/or erosions the location is important. If located at strain-related joint areas, look for other signs suggestive of degenerative disease (joint space narrowing, subchondral sclerosis, osteophytes, joint vacuum phenomenon, and subchondral cysts)

The presence of degenerative SIJ changes may not always be the cause of pain. Thus, in a CT study degenerative SIJ changes were reported to be associated with degenerative spinal changes in men, but not in women [63]. Moreover, low back pain fulfilling the definition of inflammatory back pain indicating axSpA may be due to L5-S1 pathology [17,18].

## 8. Diffuse Idiopathic Skeletal Hyperostosis (DISH)

DISH is a common condition with a prevalence above 15% in individuals >50 years of age [64]. DISH is a non-inflammatory condition characterized by pathologic calcification and ossification of entheses, primarily at the spine, but also at other joints including the SIJ [65]. Changes are often asymptomatic, but patients may present with back pain and stiffness. Thus, DISH was identified in 1.5–3.5% of the subjects referred to MRI due to clinically suspected sacroiliitis (Table 1) [17,18]. Although the etiology of DISH remains unknown, diabetes mellitus and other metabolic conditions are strongly associated with DISH.

The imaging diagnosis of DISH is mainly based on flowing ossifications along the anterior border of the spine extending over ≥4 vertebral bodies and DISH changes at the SIJ likewise present as coarse bony/ossified bridges located to the anterior and posterior SIJ ligaments and capsule (Figure 20), resembling the obliteration of the SIJ by radiography and simulating axSpA ankylosis [66,67]. Intraarticular bony projections and/or partial fusion in the upper part of the SIJ in addition to hyperostotic bridging across the joint have been observed by CT and MRI, but the lower two-thirds of the SIJ are usually spared [48,68]. Slight BME and fatty marrow metaplasia at the SIJ may occur and has been reported to be present in 16% and 30% of 38 DISH subjects, respectively [69]. Erosions and subchondral sclerosis are rare, thus differentiating the changes from axSpA involvement [66,69].

## 9. Infectious Sacroiliitis

Infectious sacroiliitis is a relatively rare condition with variable clinical and biochemical manifestations. Low back or buttock pain is characteristic, and CRP is usually elevated and often, but not always, accompanied by fever and leukocytosis [70].

The changes are mostly unilateral and extensive BME by MRI is typical, often with a sacral predominance or an even distribution, and not an iliac predominance as in axSpA [71]. There is usually concomitant joint fluid accumulation, thickened distended inflamed capsule, periarticular edema including muscle edema with or without periarticular fluid collections (Figure 21) [72]. Especially the combination of large erosions, thickened inflamed capsule, and periarticular muscle edema are suggestive of infectious sacroiliitis [71]. Abscess formation in the iliopsoas or piriformis muscle is a unique pathognomonic finding often seen in later stages of the disease and is less frequent in early disease stages [72]. In advanced stages, bony bridges, fatty replacement, and ankylosis may occur.

In contrast to infection, the typical inflammatory SIJ changes in axSpA do not cross anatomic borders and are limited to the bone and intraarticular space. In a comparison of infectious and axSpA changes, the presence of periarticular muscle edema was the only independent differentiating variable with an odds ratio of 32.9 (95% CI, 7.0–155.1) [71]. However, in rare cases axSpA, sacroiliitis can mimic infection [73], especially if elicited by previous infection (reactive arthritis).

## 10. Fractures

### 10.1. Sacral Stress Fracture

Sacral stress fracture may occur in connection with acute minor traumatic injuries and/or overuse injuries in healthy young adults with normal bones such as athletes and military recruits [74] and occasionally in connection with childbirth [9,75,76] (Figure 22). Such fractures are difficult to detect by radiography whereas MRI as well as scintigraphy are useful modalities [74,77]. MRI will typically display BME within the sacral bone on one or both sides which may be confused with BME related to axSpA sacroiliitis if it involves the peripheral subchondral areas of the sacrum. However, the fracture-related BME is mostly located within the sacral bone around the fracture line which is best visualized on T1-weighted images (Figure 22). In case of doubt interpreting the MRI, clinical and biochemical aspects can be helpful. The pain related to sacral fracture is typically exacerbated by mechanical load and improves at rest, and absence of other SpA features and negative HLA-B27 test reduce the probability of axSpA. However, a definite diagnosis of stress fracture can be established only by appropriate imaging. If the MRI is not diagnostic, a supplementary CT can show a characteristic pattern with a vertical fracture line or a sclerotic bone pattern consistent with bone bruise changes.

Apparent sacral stress fractures may sometimes be the first manifestation of osteoporosis/osteopenia and thereby represent an insufficiency fracture (Figure 23).

### 10.2. Sacral Insufficiency Fracture

Sacral insufficiency fracture is more common than stress fractures. It occurs in weakened bone, e.g., due to osteoporosis or radiotherapy, after minimal or unrecognized trauma and is mainly seen in middle-aged and elderly patients [77,78,79]. It may represent an underestimated cause of low back and pelvic pain in patients suffering from predisposing conditions, but may also occur without region-specific pain [77,79,80].

Changes by MRI consist of osseous edema, often bilateral with vertical fracture lines, best visualized on T1-weighted images (Figure 24).

Osseous edema may be pronounced, especially in previously irradiated bones and in addition to vertical fractures concomitant transverse sacral fracture may appear, best visualized on sagittal MR images, but resulting in a H-sign by bone scintigraphy and sometimes also seen on coronal MR images [80] (Figure 25). Concomitant pubic insufficiency fractures are frequent [79] and other insufficiency fractures in pelvic bones may appear (Figure 24 and Figure 25).

## 11. Inflammatory Conditions

### 11.1. Other Types of Arthritis, Including Gout and Pseudo-Gout

**Gout** is a common metabolic disease frequently affecting middle-aged men and postmenopausal women, usually with peripheral joint manifestations [81]. It may occasionally affect the axial skeleton, including the SIJ with non-specific symptoms mimicking other inflammatory conditions, although rarely detected [81]. In the literature on SIJ, gout is mainly limited to case reports [82,83,84].

The appearance by SIJ MRI is not specific although often characterized by the occurrence of intraarticular joint fluid, bony erosions filled with a material of intermediate-to-low signal intensity on T1-weighted sequences as well as on STIR/T2FS (Figure 26) and usually accompanying BME and Gadolinium enhancement in the adjacent bone marrow [84]. CT, especially dual energy CT [84], may assist in the diagnosis by displaying intraarticular mineralization and well-defined erosions.

**Calcium pyrophosphate dihydrate (CPPD)** deposition disease can involve the axial skeleton, including the SIJ [85], but may not be detected due to minor or non-specific symptom. Reports of symptomatic SIJ involvement are limited to case reports [86,87,88]. Clinically pronounced and relatively acute pain symptoms with concomitant CRP elevation can occur and simulate infectious sacroiliitis [86,87,88], especially as there in addition to subchondral BME at the SIJ can be accompanying edema in the adjacent soft tissue as in infectious sacroiliitis [87] (Figure 27). Such SIJ changes are comparable with the pronounced active inflammatory CPPD changes sometimes seen in the spine [89,90].

### 11.2. Chronic Non-Bacterial Osteitis (CNO)

CNO encompasses a spectrum of relatively rare auto-inflammatory conditions sharing musculoskeletal and cutaneous manifestations [91,92]. It occurs at all ages, but usually starts before the age of 60 years. In children/adolescents, the condition has often been described under the terms **CRMO** (chronic recurrent multifocal osteomyelitis) whereas the term **SAPHO** (Synovitis, Acne, Pustulosis, Hyperostosis) has frequently been used in adults. CNO is characterized by non-infectious chronic osteitis, involving both the cortical area and the bone marrow with characteristic endosteal as well as periosteal new bone formation (Figure 28 and Figure 29). Symptoms usually fluctuate with flares of pain during periods with active lesions declining or vanishing during quiet periods. Most patients present with polyostotic involvement often accompanied by skin lesions, most frequent pustulosis palmoplantaris [91].

The site of disease involvement is age-related. In children, the metaphysis of long tubular bones, the spine and clavicles are most commonly involved [93] whereas the sterno-costo-clavicular region is the most common location in adults, followed by the spine and SIJ [91,92,94]. CNO in childhood may occasionally persist in adulthood with a change in the location of lesions.

CNO changes at the SIJ are most frequent in adults where they are often unilateral and confined to one of the bones, most common the iliac bone [92]. The MRI appearance depends on the stage of disease. During active stages, there is BME and enhancement whereas chronic changes are predominantly characterized by sclerosis and hyperostosis with concomitant fat metaplasia in the bone marrow corresponding to previous BME areas. The diagnosis is often based on a combination of clinical and radiologic findings, but a supplementary biopsy may be needed in case of tumor-suspected lesions (Figure 28).

In childhood, CNO lesions at the SIJ will often be part of a multifocal osseous disease with accompanying lesions in other pelvic bones (Figure 29) or in other skeletal areas. Whole body MRI is therefore a preferred examination in children/adolescents to detect symptomatic as well as clinically silent lesions [95].

## 12. Hyperparathyroidism and Other Disorders of Mineral Metabolism

**Hyperparathyroidism**, either primary or secondary to renal failure, causes increased bone remodeling with prevailing bone resorption [96]. Subchondral bone resorption can be seen in synchondrosis joints such as the sacroiliac and sternoclavicular joints as well as the pubic symphysis [97]. Changes at the SIJ usually present as bilateral and symmetrical irregularities of the iliac joint margins with gross erosion-like lesions and pseudo-widening of the joint due to bone resorption, changes mimicking structural lesions in axSpA sacroiliitis [98,99,100]. There may be subchondral BME which in combination with pseudo-widening of the joints can simulate active axSpA changes [98] (Figure 30). However, joint space narrowing or bone bridging/ankylosis and sacral-sided erosions are not features of changes caused by hyperparathyroidism [99].

**Hypoparathyroidism,** a metabolic bone disorder often caused by iatrogenic injury during neck surgery, is characterized by reduced bone remodeling resulting in increased osseous mineralization [96,100] and occasionally by ligamentous ossification, which can simulate paravertebral ossification in DISH and axSpA, especially psoriatic SpA. The SIJ is generally spared, but ligamentous changes may occur simulating those observed in DISH.

In the differentiation of metabolic mineral disorders from axSpA it is important that clinical, biochemical, and imaging findings are all taken into account, especially biochemical findings of normal C-reactive protein and negative HLA B27, abnormal level of serum phosphate, elevated alkaline phosphatase, low serum 1,25-dihydroxyvitamin D levels, and/or a history of renal failure [48,101].

## 13. Tumors

The pelvic bones are common sites for metastasis and myeloma giving rise to pain, but the lesions will rarely be located directly adjacent to the SIJ and simulate sacroiliitis. The rare occurrence of primary mesenchymal tumors such as osteochondromas, giant cell tumors, and sarcomas (Figure 31) will usually imply an osseous or soft tissue extension and rarely simulate sacroiliitis [102]. Sacral meningeal cysts (Tarlov cysts) can be the cause of low back pain, especially when voluminous, but are confined to the sacral canal or foramina and do thus not simulate sacroiliitis by imaging.

## 14. Conclusions

In the diagnosis of axSpA sacroiliitis, it is important to be aware of differential diagnoses simulating sacroiliitis by MRI. Moreover, the diagnosis of axSpA sacroiliitis should not only be based on imaging findings. To obtain a correct diagnosis of axSpA, especially in cases of doubt, a multidisciplinary approach by rheumatologists and radiologists is necessary to ensure that both clinical, laboratory, and imaging findings support the diagnosis.

## Figures and Tables

**Figure 1 jcm-12-01039-f001:**
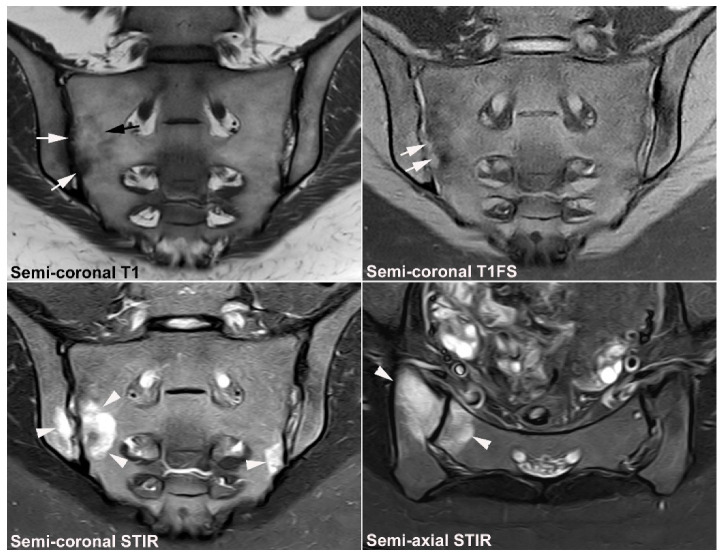
**Predominant active sacroiliitis changes** in a 31-year-old woman with inflammatory low back pain for 8 months, normal radiographic findings, CRP 22 mg/L (normal < 8 mg/L) and HLA B27 negative. Upper row of images: semi-coronal T1 and T1FS; lower row: semi-coronal STIR and semi-axial STIR images. There is manifest subchondral BME at the inferior part of the joints, especially on the right side (arrowheads), in addition to right-sided sacral erosions (white arrows) with an adjacent area of faint subchondral fat deposition (black arrow).

**Figure 2 jcm-12-01039-f002:**
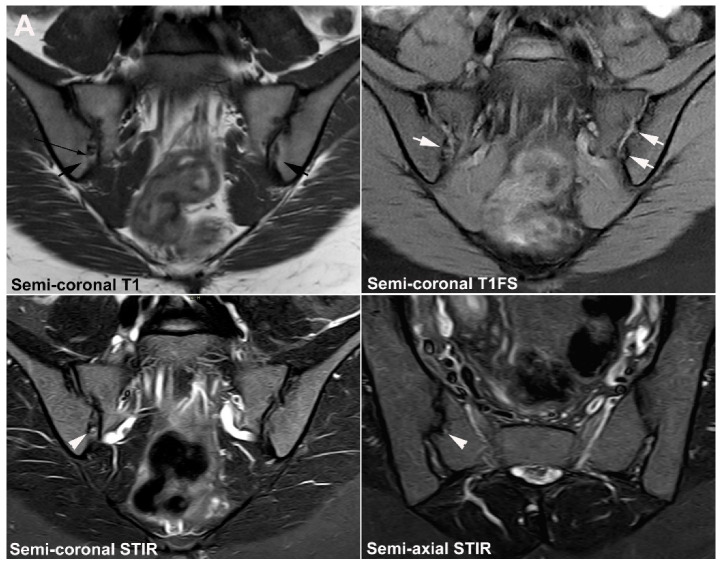
**Predominant structural sacroiliitis changes** in a 22-year-old woman with intermittent inflammatory low back pain for 2 years and pronounced hip pain for two months, equivocal radiographic SIJ findings, CRP 65 mg/L (normal < 8 mg/L) and HLA B27-positive. (**A**) Upper row of images: semi-coronal T1 and T1FS; lower row: semi-coronal STIR and semi-axial STIR image at the middle portion of the joints. There are bilateral erosions (white arrows) in addition to fat deposition (black arrows), including a small fat deposition in an erosive cavity (backfill) in the right ileum (thin black arrow). There is no manifest subchondral BME, but edema in two erosion cavities (arrowheads). (**B**) The inferior semi-axial slices of the SIJ (left image) revealed bilateral hip synovitis and a supplementary pelvic STIR sequence (right image) shows manifest bilateral active erosive hip arthritis.

**Figure 3 jcm-12-01039-f003:**
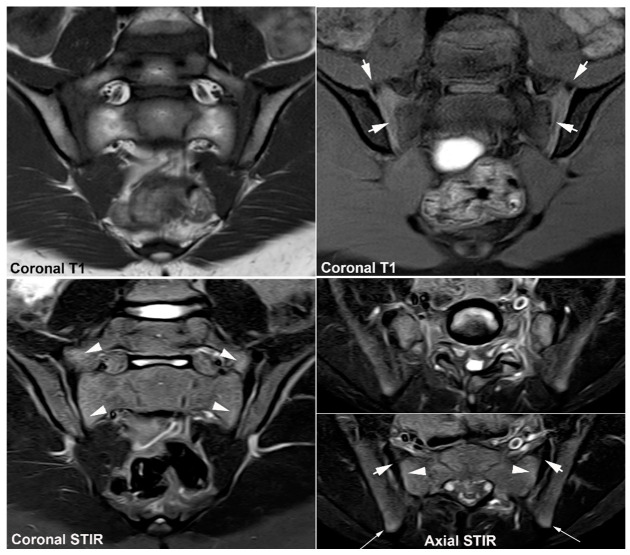
**Normal SIJ in a 10-year-old boy.** Upper row of images: semi-coronal T1 and T1FS; lower row: semi-coronal STIR and two semi-axial STIR images, superiorly at S1 (upper images) and at the border between S1 and S2, respectively. There is a brim of subchondral edema in the sacrum (arrowheads) nearly as bright as the edema at the iliac crest posteriorly (thin long arrows). Moreover, persistence of intervertebral space between S1 and S2, and visible nuclei at the sacral wings and in the joint spaces, best visualized on the T1FS image and on the axial slice at the border between S1 and S2 (arrows).

**Figure 4 jcm-12-01039-f004:**
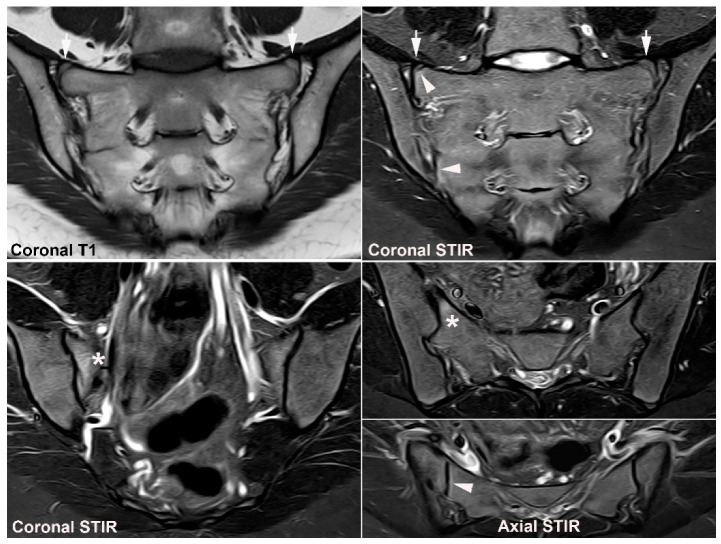
**Normal SIJ in a 17-year-old girl.** Upper row of images: semi-coronal T1 and STIR images posteriorly at the sacral wings; lower row: semi-coronal STIR more anteriorly at the border between S1 and S2, and two semi-axial STIR images, one at the level of S1/S2 (upper image) and one at the inferior part of S2. There is still a symmetrical subchondral brim of edema in the sacrum (arrowheads) in addition to persistence of unfused nuclei at the sacral wing (arrows) with a brim of adjacent edema, most pronounced at the right side (arrowhead). The nuclei between S1 and S2 have fused with the sacrum, but there is persistence of edema adjacent to the nucleus area on the right side (asterisk), which must not be mistaken for inflammatory changes.

**Figure 5 jcm-12-01039-f005:**
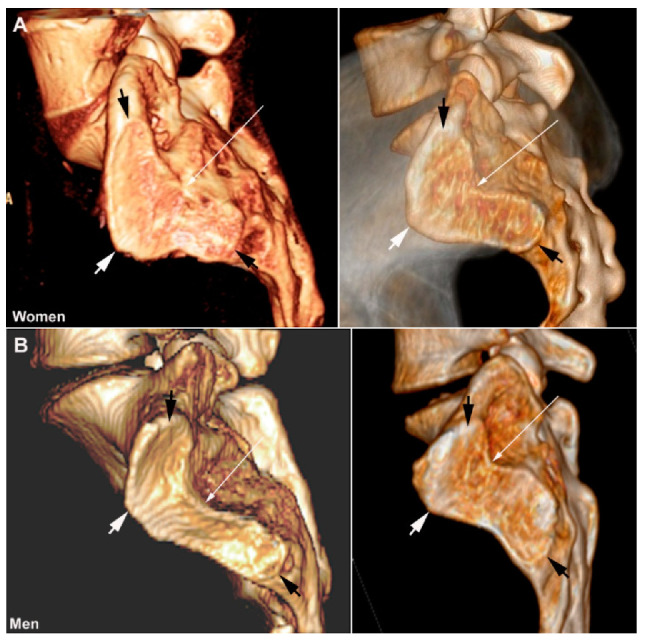
**Normal joints,** 3D CT reconstructions of the sacral joint facets showing the variable form and sex differences with a more horizontally located joint facet in women (**A**) than in men (**B**). The joint facet in women is usually broadest at the upper and middle portion of the joint whereas the width of the joint facet is less variable in men. The cartilaginous joint facets are marked with black arrow superiorly and inferiorly and with white arrows on the most anterior portion and thin white arrows at the deepest posterior point at the transition between the cartilaginous and ligamentous joint portion.

**Figure 6 jcm-12-01039-f006:**
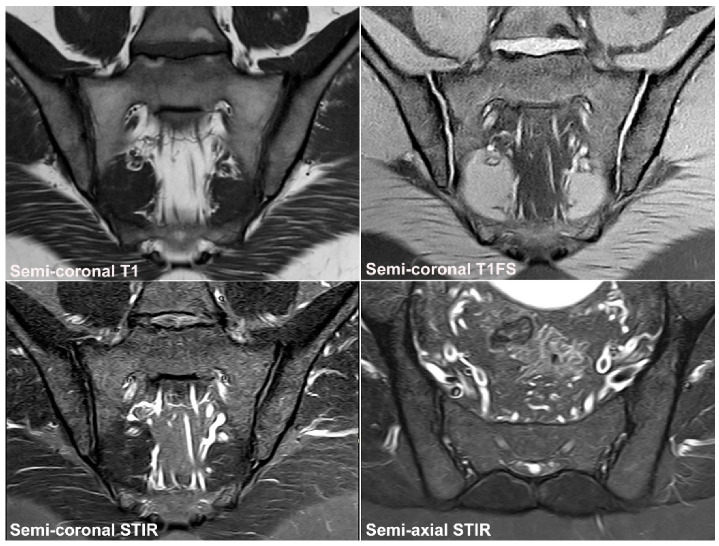
**Recommended standard for a diagnostic SIJ MRI.** The examination should include at least four sequences: A semi-coronal T1 and a cartilage sequence such as T1FS or a gradient echo sequence and STIR or T2FS in two perpendicular planes (semi-coronal and semi-axial) [34,35,36].

**Figure 7 jcm-12-01039-f007:**
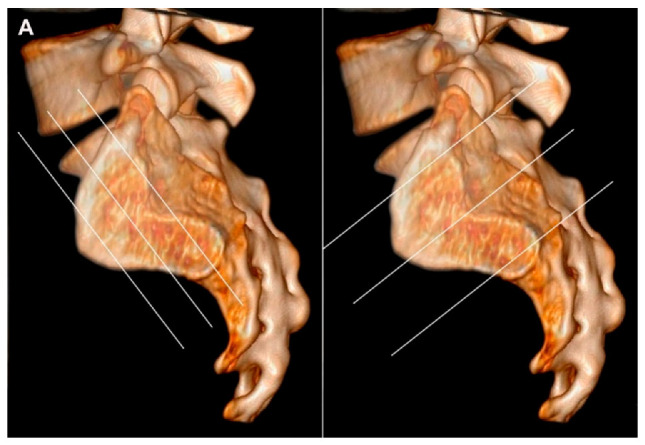
(**A**) Three-dimensional CT reconstruction of the sacral joint facet with lines corresponding to the two perpendicular scan planes, semi-coronal and semi-axial slice orientation, respectively. The semi-coronal slice orientation should be standardized to align the posterior margin of the second sacral vertebra or be perpendicular to the vertebral surface of S1. The scan volume should include the entire cartilaginous and ligamentous joint compartment, often implying slices outside the joint areas. (**B**) MRI of normal SIJs in a 31-year-old man with disk degeneration, three semi-coronal T1-weighted images, anterior, in the middle and posterior in the joint, and three semi-axial STIR images, at S1, S2, and S3 respectively, corresponding to the slice positions indicated on A and showing the anatomical localization of the different joint areas on the two perpendicular slice orientations. On the axial slices, the cartilaginous joint portion is marked with asterisks at the sacral side and the ligamentous portion with # at the iliac side of the joint. The axial image at S3 illustrate the use of a large field of view for semi-axial slices visualizing the upper part of the hips and the symphysis.

**Figure 8 jcm-12-01039-f008:**
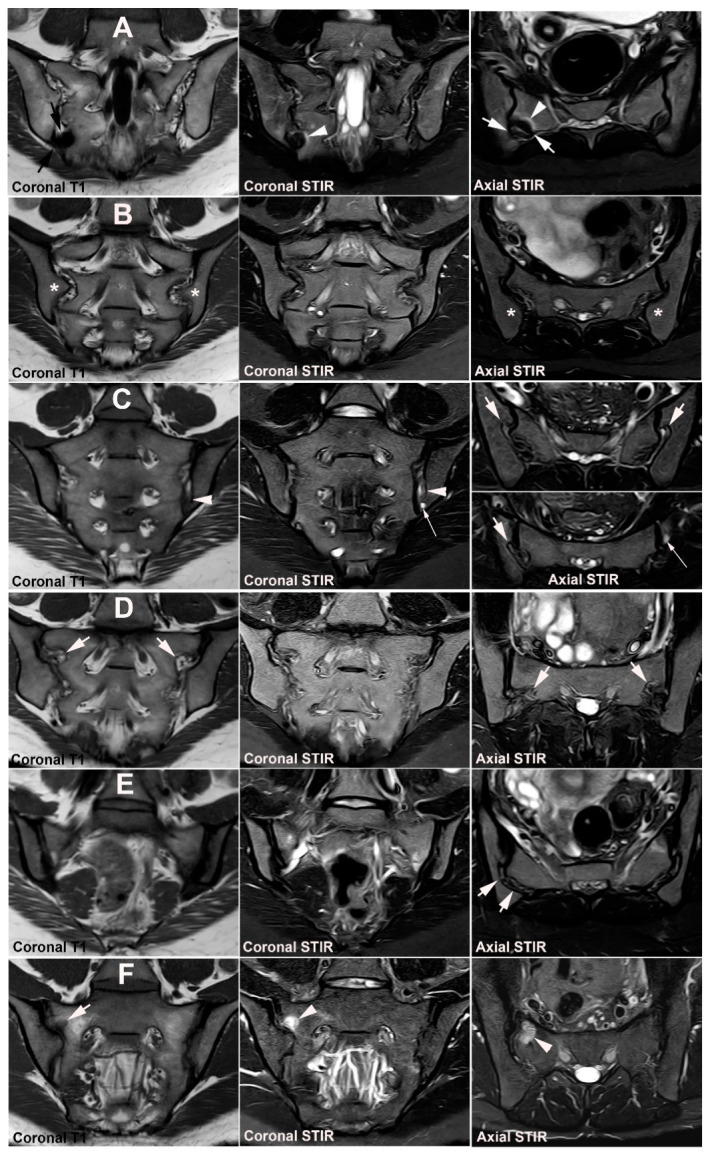
**Anatomical SIJ variation.** The figures illustrate the spectrum of SIJ variations, except unfused sacral nuclei illustrated separetely. Semi-coronal T1-weighted images to the left, semi-coronal STIR images in the middle, and semi-axial STIR images to the right. (**A**)**: *Accessory (ACC) SIJ*** is an additional small joint in the ligamentous joint compartment with opposing articular joint facets. It can be bilateral or unilateral and is most frequent at the level of S2 but can also occur at S1 and S3 [25,46]. ACC SIJ is best visualized on semi-axial slices but can be confirmed in the semi-coronal plane by the lack of interposed fat and ligaments between the joint facets (black arrows). In this patient presenting with pain, there is concomitant subchondral sclerosis at the ACC joint (white arrows) surrounded by a brim of osseous edema (arrowheads). (**B**)**: *Iliosacral complex*** consists of an osseous protrusion of the iliac bone in the ligamentous joint compartment (asterisks) with a corresponding groove in the opposing sacral bone visualized in both the semi-coronal and semi-axial plane. (**C**)**: *Bipartite iliac bony plate*** is a cleft-like appearance in the posterior/inferior part of the iliac bone best visualized in the semi-axial plane (arrows). It is formed by a deep insertion of a posterior iliosacral ligament [30] which is often surrounded by vessels causing edematous changes as seen on the semi-coronal and inferior semi-axial STIR images (long thin arrow). On semi-coronal images the osseous defect containing the ligament may simulate erosion (arrowheads). (**D**)**: *Semicircular defects*** are round well-defined defects in the iliac and/or sacral bone superiorly in the ligamentous joint compartment (arrows). (**E**)**: *Crescent-like iliac bony plate*** is a concave configuration of the iliac bone posteriorly and superiorly in the ligamentous joint compartment (arrows) with a congruent bulging of the opposite sacral bone. It can be difficult to detect by MRI using standard semi-axial slice orientation [40]. (**F**)**: *Dysmorphic SIJ*** is a term used for manifest osseous protrusions in the cartilaginous joint compartment, either of the sacrum or the ilium [40,45]. It can be accompanied by strain-related subchondral edema (arrowheads) and/or erosion-like osseous irregularities (arrow) and thereby mimic sacroiliitis as shown in Figure 9.

**Figure 9 jcm-12-01039-f009:**
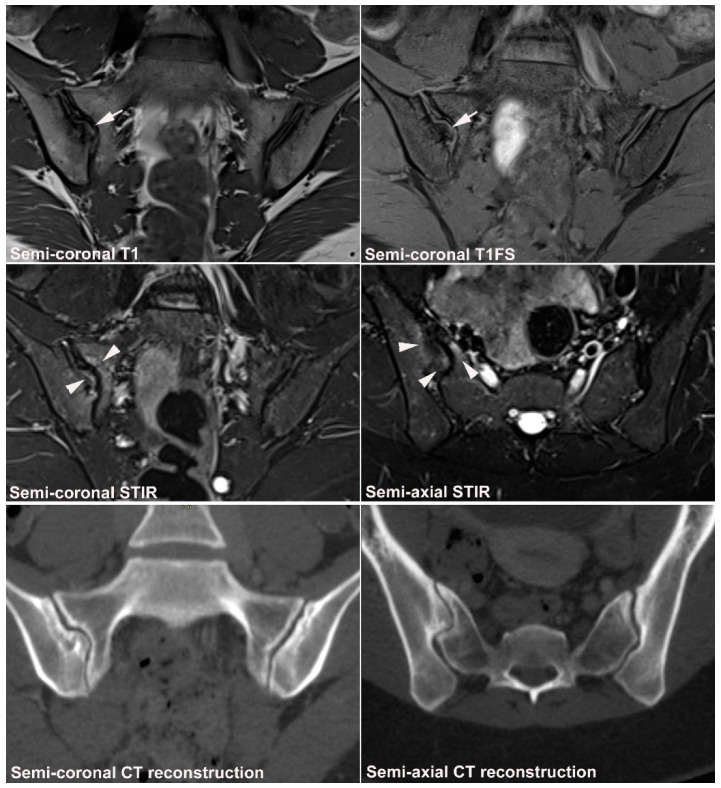
**Dysmorphic SIJ** in a 30-year-old woman with recurrent low back and right-sided buttock pain for three years. Semi-coronal T1, T1FS and STIR, and semi-axial STIR images with supplementary CT reconstructions in a semi-coronal and semi-axial orientation, respectively (lower row of images). There is manifest osseous protrusion of the right ileum into the sacrum with subchondral edema (arrowheads) and erosion-like irregularity at the iliac joint facet (arrows). The CT images clearly show the dysmorphic changes and absence of erosion.

**Figure 10 jcm-12-01039-f010:**
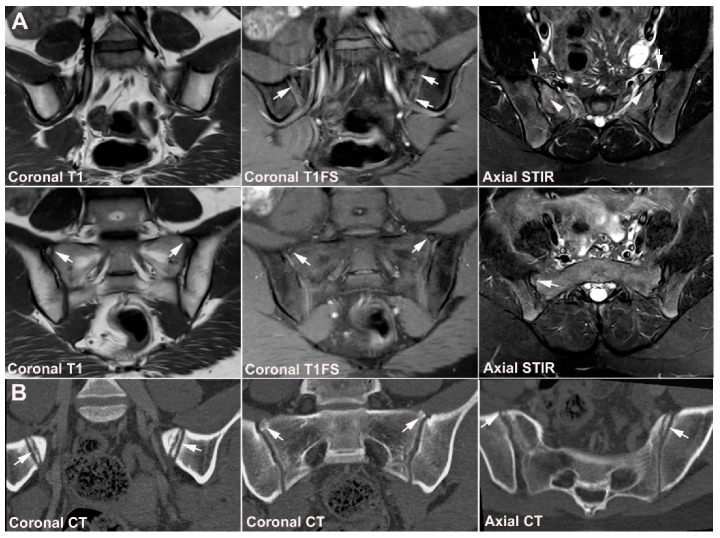
**Unfused nucleus at the sacral wings and at the border between S1 and S2 in adulthood.** (**A**) MRI in a 20-year-old healthy woman [40], semi-coronal T1 and T1FS, and semi-axial STIR images, upper image row anteriorly at the border between S1 and S2 and lower row more posteriorly in the joint at S1. There is persistence of small nuclei at the sacral wings posteriorly and at the border between S1 and S2 anteriorly (arrows) causing subchondral edema, especially at the S1/S2 region (arrowhead). (**B**) CT images obtained one year previously, semi-coronal and semi-axial reconstructions more clearly show the unfused nuclei both at the sacral wings and in the joint space (arrows).

**Figure 11 jcm-12-01039-f011:**
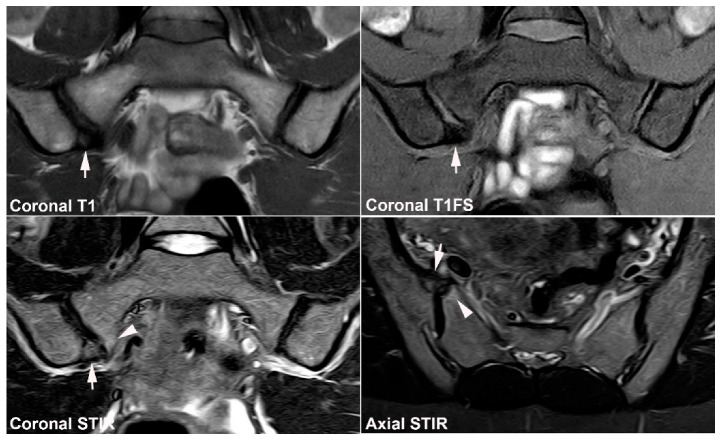
**Unfused nucleus at the border between S1 and S2.** MRI in a 30-year-old woman with intermittent low back pain; semi-coronal T1, T1FS and STIR, and semi-axial STIR images at the border between S1 and S2. There is persistence of a nucleus in the right joint space (arrows) accompanied by subchondral edema in the adjacent sacral bone (arrowheads).

**Figure 12 jcm-12-01039-f012:**
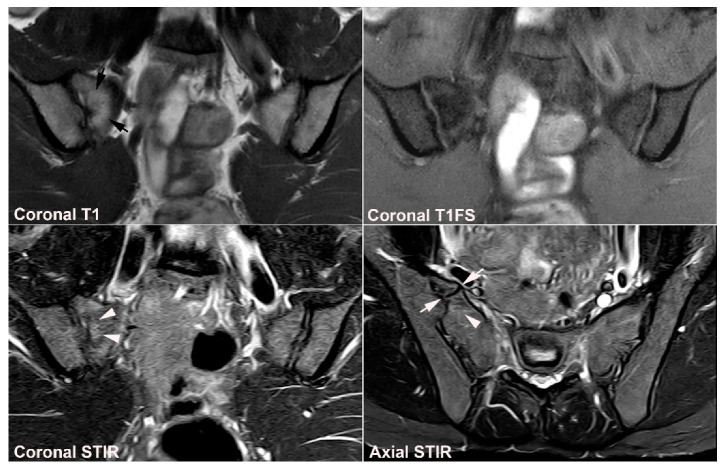
**Atypical fusion of nucleus** in a 24-year-old man with right-sided buttock pain for 8 months. Semi-coronal T1, T1FS and STIR, and semi-axial STIR images at the border between S1 and S2. At the right SIJ there is a relative hypertrophic nucleus at the border between S1 and S2 which has fused with the ileum and not as normally with the sacral bone (white arrows). It causes an indentation in the sacral bone with a surrounding brim of edema (arrowheads), focal irregular joint margins and fat deposition (black arrows on T1).

**Figure 13 jcm-12-01039-f013:**
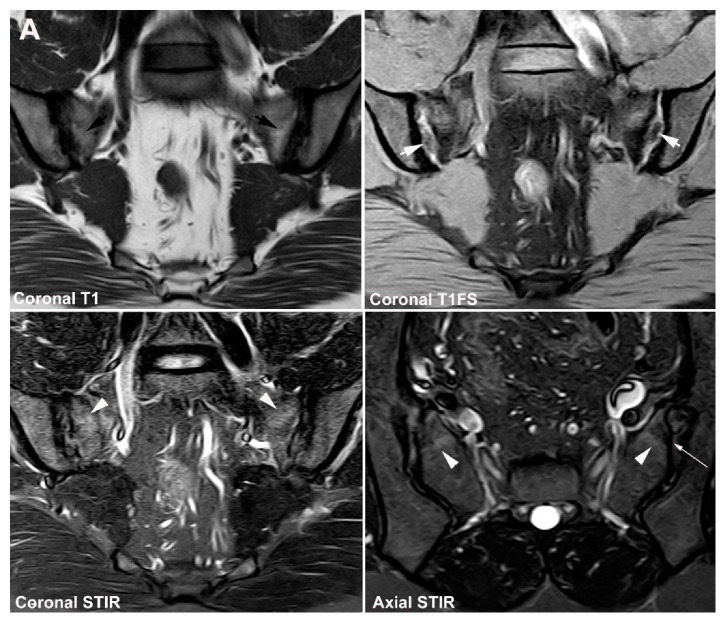
**Intraarticular unfused nuclei.** (**A**) MRI in a 30-year-old sports-active man with intermittent low back and buttock pain for 5 years, semi-coronal T1, T1FS, and STIR, and semi-axial STIR images show irregular joint facets with dispersed subchondral edema (white arrowheads) most pronounced adjacent to the previous sacral nucleus area at the border between S1 and S2, best seen on the axial slices (long thin arrow). There is additional dispersed subchondral fat deposition (black arrows on T1) and condensed osseous structures in the joint space on the T1FS image (white arrows), most obvious on the left side. (**B**) Supplementary CT, semi-coronal, and semi-axial reconstruction confirmed the presence of persistent nuclei in the joint space in addition to degenerative changes with osteophyte formation (arrow), but no erosions characteristic of axSpA sacroiliitis.

**Figure 14 jcm-12-01039-f014:**
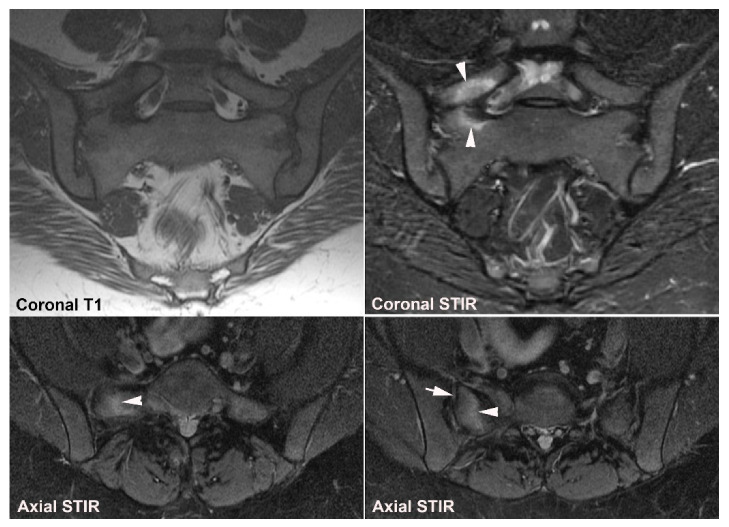
**Transitional vertebra,** MRI in a 33-year-old man with inflammatory right-sided low back pain for 5 months. Semi-coronal T1 and STIR and two semi-axial STIR images, at the level of the transitional vertebra and below in the sacrum. There is lumbarization of S1 with enlarged transverse processes articulating with the upper border of the sacrum (Castellvi type IIB) and pronounced BME around the right-sided pseudo-articulation (arrowheads). On the axial slice at the upper part of the sacrum, the BME reaches the subchondral SIJ area (arrow), but there are no structural changes.

**Figure 15 jcm-12-01039-f015:**
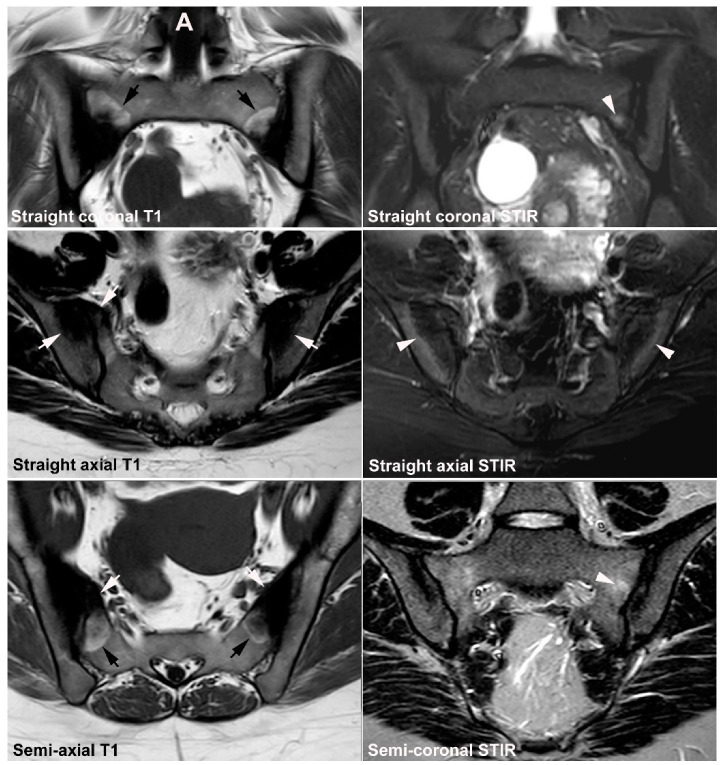
**Osteitis condensans ilii (OCI).** (**A**) MRI of a 32-year-old woman with three preceding pregnancies, straight coronal and axial T1 and STIR images (two upper image rows) in addition to semi-axial T1 and semi-coronal STIR of the SIJ. There is manifest bilateral iliac sclerosis in addition to sclerosis anteriorly in the sacrum (white arrows) with surrounding pronounced fat deposition anteriorly in the sacrum (black arrows on T1). There is a brim of BME peripheral to the iliac sclerosis in addition to anteriorly in the sacrum on the left side (arrowheads). (**B**) Pelvic radiograph shows typical OCI changes in the form of bilateral iliac sclerosis (asterisks) in addition to para-glenoidal sulci (arrows) consistent with strain-related changes.

**Figure 16 jcm-12-01039-f016:**
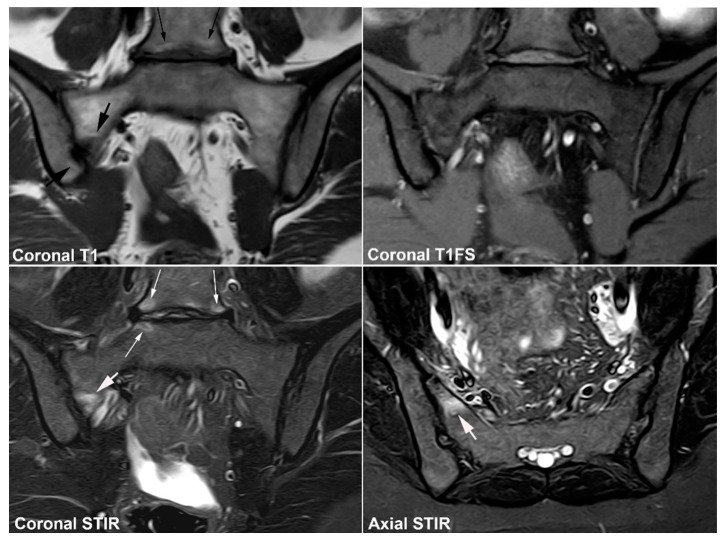
**Postpartum BME** in a 38-year-old woman complaining of persistent low back pain 12 months after a childbirth, semi-coronal T1, T1FS and STIR, and semi-axial STIR images showing subchondral BME at the anterior portion of the right SIJ (white arrows) with a brim of surrounding fat deposition (black arrows on T1). There are additional disk degenerative changes with subchondral BME and fat deposition (long thin white and black arrows, respectively).

**Figure 17 jcm-12-01039-f017:**
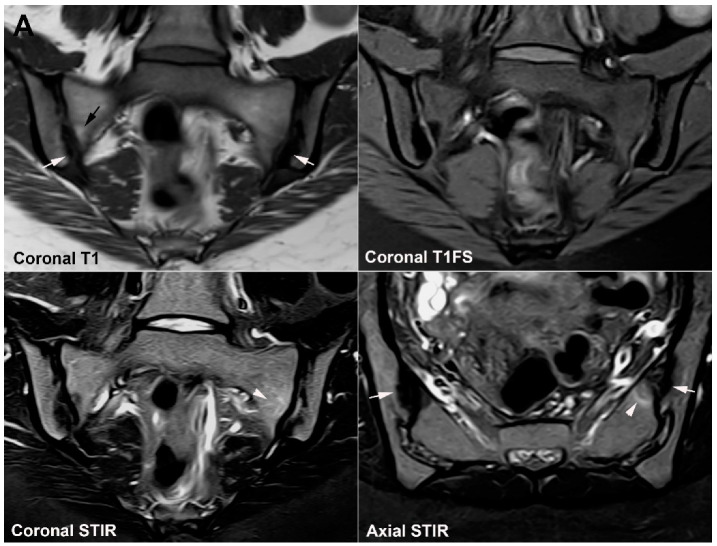
**Evolution of postpartum MRI changes to OCI.** MRI in a 26-year-old woman (**A**) 3 and (**B**) 12 months after her first normal vaginal delivery, semi-coronal T1, T1FS and STIR, and semi-axial STIR images at both time points. Iliac sclerosis was present at both time points (white arrows) but became more homogeneous and signal void during the 9-month period. At 3 months postpartum there was concomitant BME at the anterior middle part of the sacrum on the left side (arrowheads) which vanished 12 months postpartum. Small areas with fat deposition (black arrows on T1) either located in the subchondral bone or peripheral to the iliac sclerosis were seen at both-time points, but no erosions.

**Figure 18 jcm-12-01039-f018:**
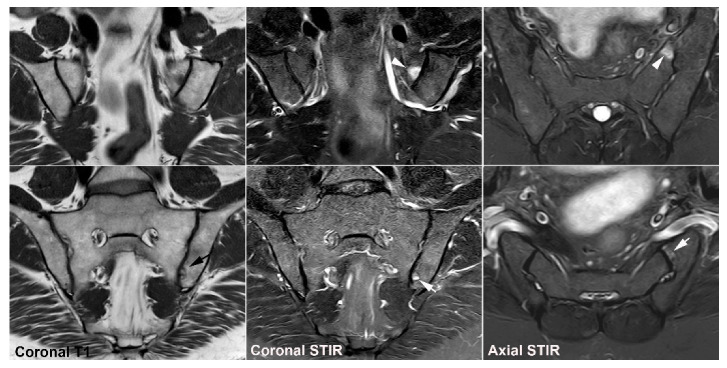
**Strain-related SIJ changes.** MRI in a 53-year-old sports active man, semi-coronal T1 to the left, semi-coronal STIR in the middle, and semi-axial STIR to the right with the upper images row displaying the anterior middle part and the lower image row the inferior part of the joints. There is subchondral BME anteriorly in the sacrum corresponding to the strain-related joint area (arrowheads) and a subchondral cyst with fluid signal inferiorly in the left ileum with slight surrounding fat deposition (black arrow on T1); no erosions or fat deposition characteristic of axSpA.

**Figure 19 jcm-12-01039-f019:**
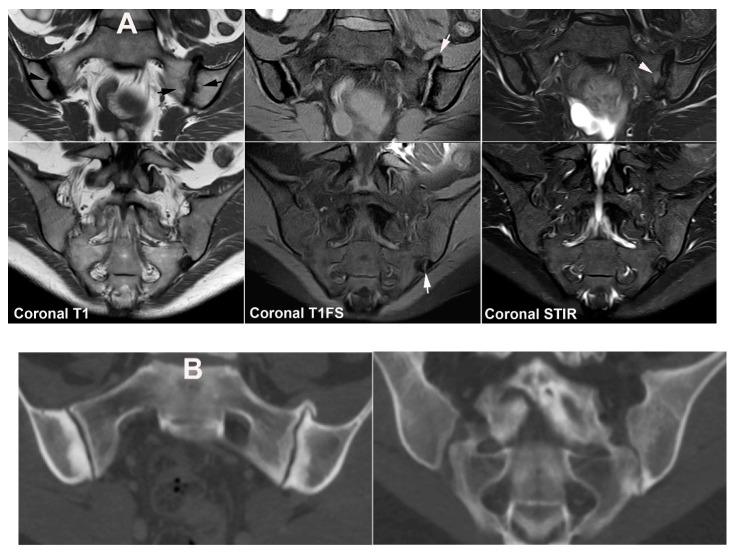
**Degenerative SIJ changes.** (**A**) MRI in a 39-year-old woman with scoliosis complaining of persistent pain with inflammatory characteristics in the region of the SIJs, especially the left SIJ, semi-coronal T1, T1FS, and STIR at the anterior part of the joint (upper image row) and posteriorly (lower image row) showing irregular joint surfaces with subchondral sclerosis anteriorly and dispersed fat deposition (black arrows on T1) in addition to osteophyte formation superiorly and inferiorly at the left SIJ (white arrows). There is also a small area of subchondral BME at the left SIJ (arrowhead). (**B**) A supplementary CT performed some months later related to scoliosis operation, semi-coronal reconstructions anteriorly and posteriorly corresponding to the MR images confirmed all degenerative changes and revealed no erosions.

**Figure 20 jcm-12-01039-f020:**
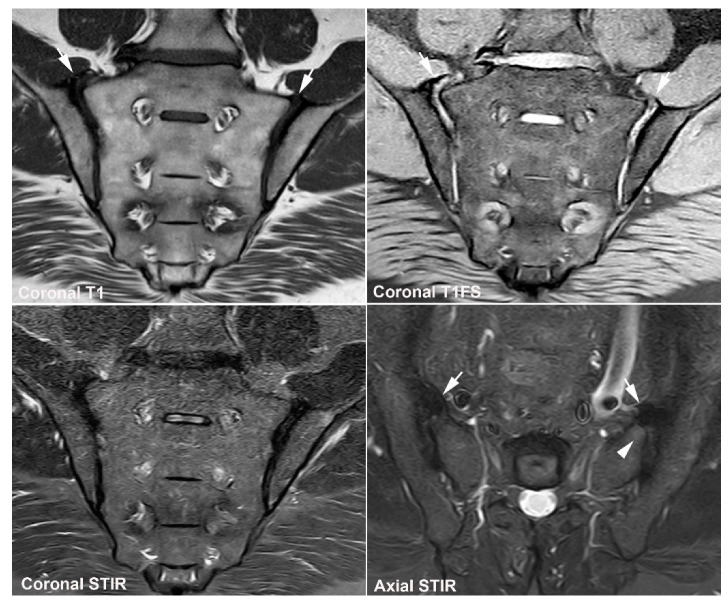
**Diffuse idiopathic skeletal hyperostosis (DISH).** MRI in a 64-year-old man presenting with flowing thoracic paravertebral ossification conforming to DISH, semi-coronal T1, T1FS, and STIR, and semi-axial STIR images of the SIJ showing periarticular new bone formation superiorly and anteriorly at the joints (arrows) in addition to a slight subchondral BME in the sacrum on the left side (arrowhead), but no manifest BME or erosions.

**Figure 21 jcm-12-01039-f021:**
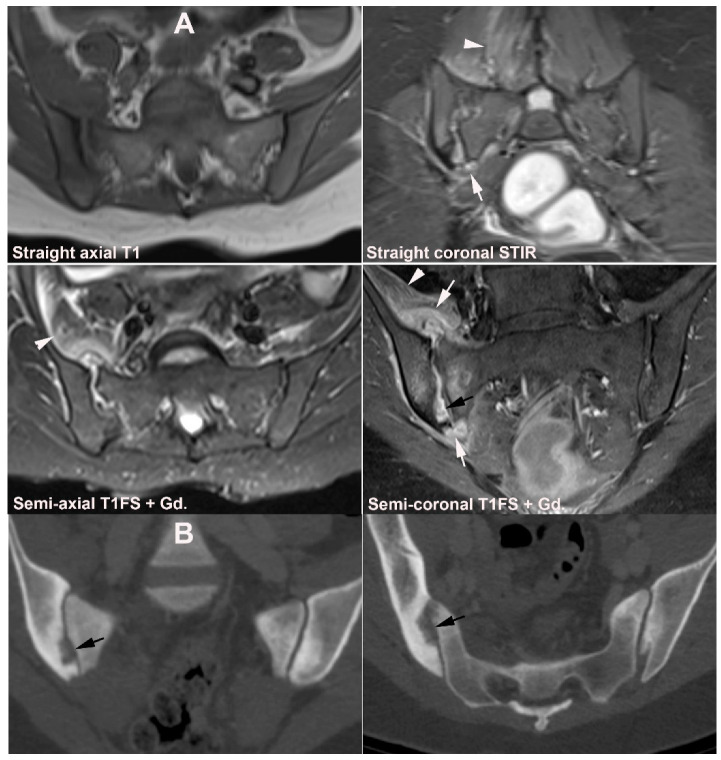
**Infectious sacroiliitis.** (**A**) MRI in a 30-year-old woman presenting with buttock pain for three weeks accompanied by malaise and fever, straight axial T1 and coronal STIR, and postcontrast semi-axial and semi-coronal T1FS. There are manifest edema and enhancement in the right joint space accompanied by subchondral edema and enhancement in addition to a thickened edematous and enhancing joint capsule (white arrows) and edema/enhancement in the surrounding soft tissue, especially in the right iliopsoas muscle (arrowheads). There seems to be an osseous defect inferiorly in the ileum (black arrow). (**B**) Supplementary CT, semi-coronal, and semi-axial reconstruction confirmed the presence of osseous destruction (black arrows), but no signs of osseous sequestration.

**Figure 22 jcm-12-01039-f022:**
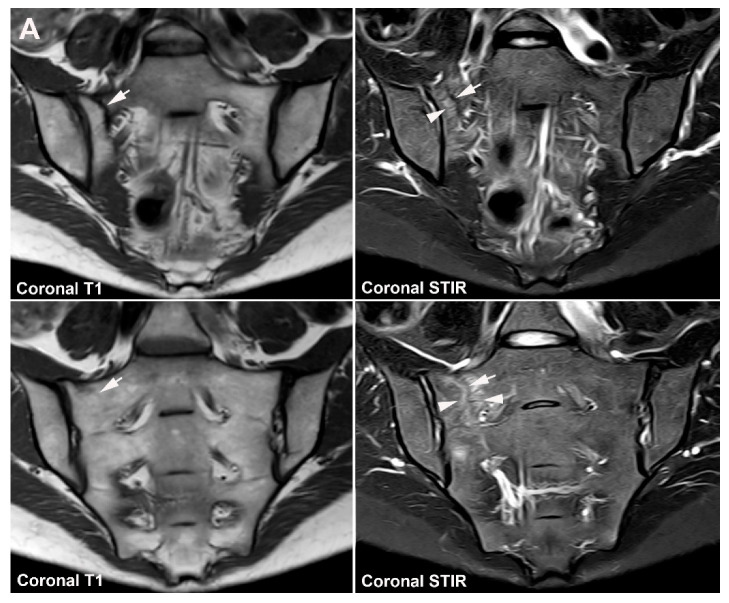
**Sacral stress fracture.** (**A**) MRI of the SIJ in a 32-year-old woman with pelvic pain during 3 months after a protracted childbirth, semi-coronal T1 and STIR images anteriorly (upper image row) and in the middle of the joint (lower image row) showing an area with BME (arrowheads) around a sclerotic line (arrows) consistent with a fracture line which is most clearly delineated on the T1-weighted image anteriorly. (**B**) Supplementary MRI of the symphysis, semi-coronal T1 and STIR, and semi-axial T2 and T2FS revealed a concomitant fracture in the right pubic bone (arrows) with surrounding BME (arrowheads).

**Figure 23 jcm-12-01039-f023:**
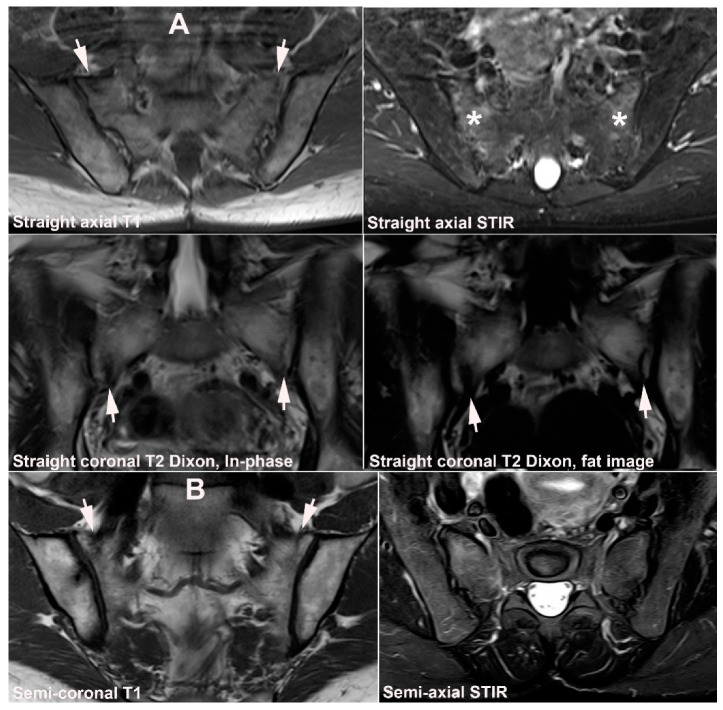
**Sacral stress fracture in a patient with osteopenia.** (**A**) MRI in a 57-year-old woman with low back pain after jogging, straight axial T1 and STIR, and coronal T2 mDixon with In-phase (T2) and fat image showing diffuse BME in the lateral part of the sacrum reaching the subchondral areas (asterisks) with visible fracture lines in the middle anterior portion of the sacrum on the T1-weighted and the Dixon images (white arrows). (**B**) One month later a dedicated SIJ MRI was performed, semi-coronal T1 and semi-axial STIR. The BME had regressed, but there was still subchondral edema, especially on the right side, and visible fracture lines on the T1-weighted image (arrows). Subsequent DEXA scanning revealed osteopenia, indicating a combination of sacral stress and insufficiency fracture.

**Figure 24 jcm-12-01039-f024:**
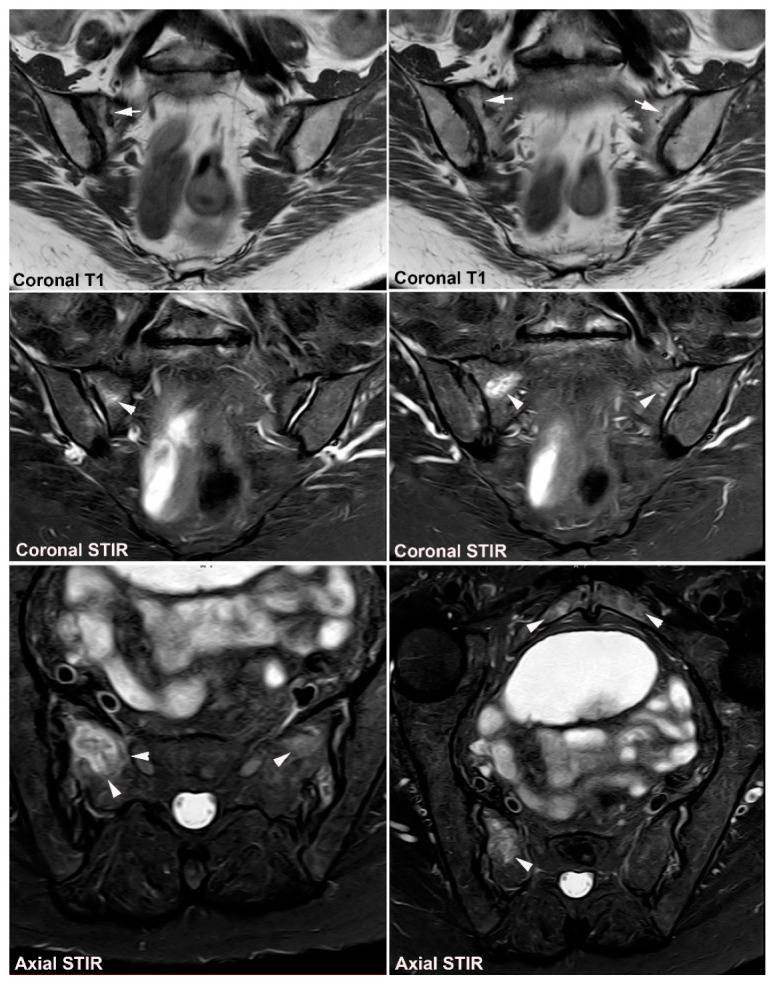
**Sacral insufficiency fracture in osteoporosis.** MRI in a 77-year-old woman with a Th12 osteoporotic fracture and low back pain. Two semi-coronal T1-weighted images (upper image row) and corresponding semi-coronal STIR images (middle image row) anteriorly in the joints in addition to semi-axial STIR images, one at the upper part and one at the middle part of the joint including the pubic bone. There is subchondral BME at the right SIJ (arrowheads) and an obvious fracture line is seen on the right side on the T1-weighted images in addition to a faint sclerotic line on the left side (arrows) with concomitant surrounding edema, best visualized on the axial images (arrowheads). The axial STIR image reveals concomitant bilateral subchondral BME at the symphysis (arrowheads) due to concomitant pubic fractures or bone bruise changes.

**Figure 25 jcm-12-01039-f025:**
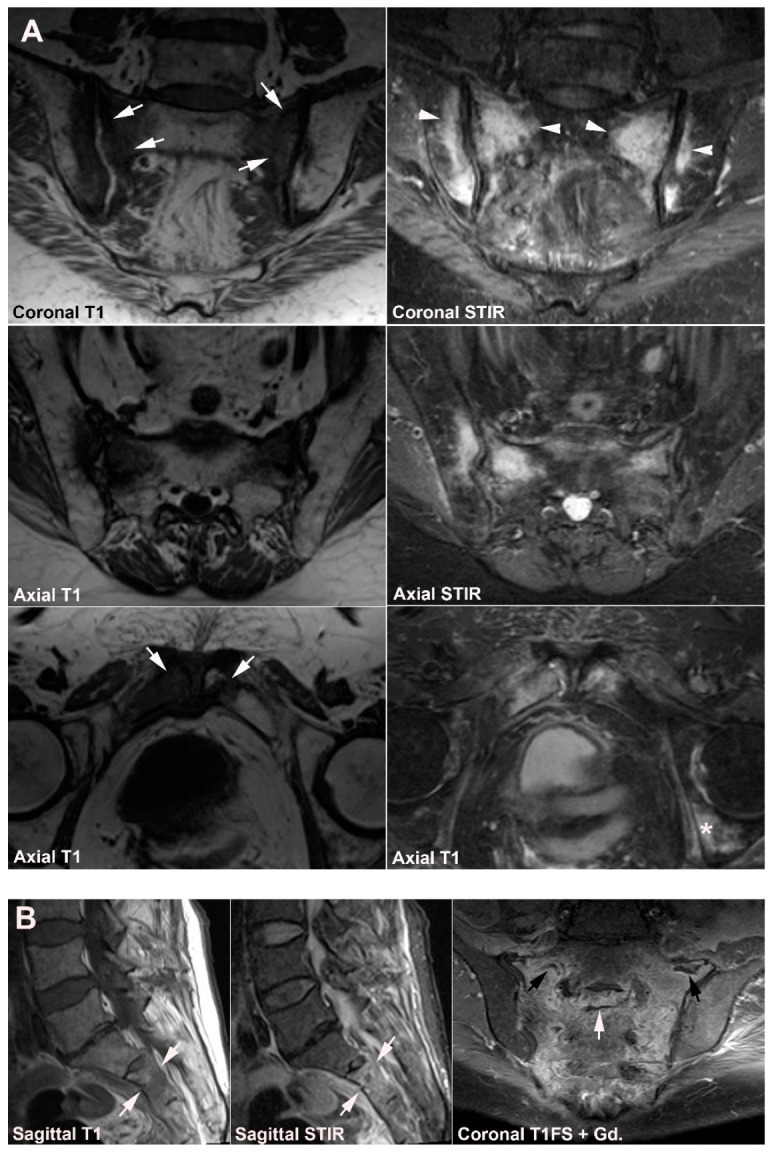
**Insufficiency fracture after pelvic irradiation** due to gynecological malignancy. (**A**) MRI in an 82-year-old woman with low back and left-sided hip pain, semi-coronal and semi-axial T1 and STIR of the SIJ (two upper image rows) in addition to semi-axial T1 and STIR of the symphysis (lower image row) showing bilateral fracture lines in the sacrum (arrows) with pronounced surrounding BME and subchondral BME in the ileum, most pronounced on the right side (arrowheads). There is a concomitant bilateral fracture in the pubic bones likewise with pronounced surrounding edema and BME posteriorly at the acetabulum (asterisk) indicating an insufficiency fracture. (**B**) MRI in a 75-year-old woman, sagittal T1 and STIR of the sacrum and postcontrast semi-coronal T1FS showing pronounced a diffuse enhancement of the sacrum with fracture lines laterally (black arrows) in addition to a transverse fracture in the upper part of S2 visible on the coronal image, but more easily detectable on the sagittal images (white arrows).

**Figure 26 jcm-12-01039-f026:**
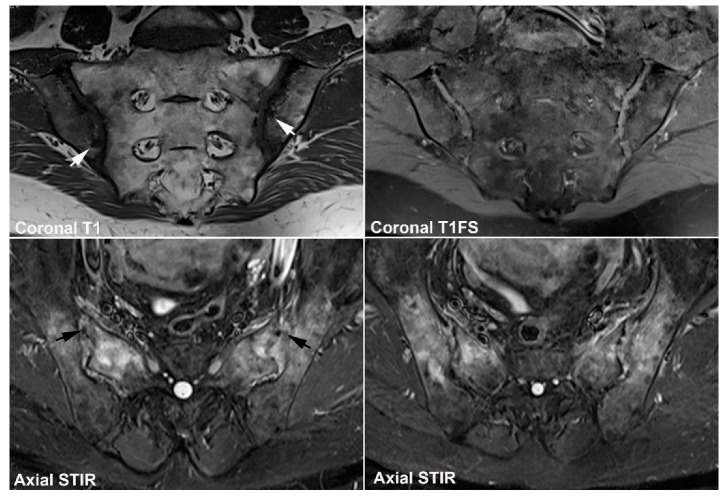
**Gout changes** in a 71-year-old woman with low back pain for approx. 12 months. MRI, semi-coronal T1 and T1FS, and two semi-axial STIR images in the middle of the joints showing relatively well-defined erosions filled with a material of intermediate-to-low signal intensity on T1 with dispersed surrounding sclerosis and small fat depositions (white arrows). The STIR images show inhomogeneous edema in the joint space with intraarticular small signal void areas (black arrows) and there is surrounding pronounced BME involving both the sacrum and the ileum. Biopsy confirmed the presence of crystals in the erosive area.

**Figure 27 jcm-12-01039-f027:**
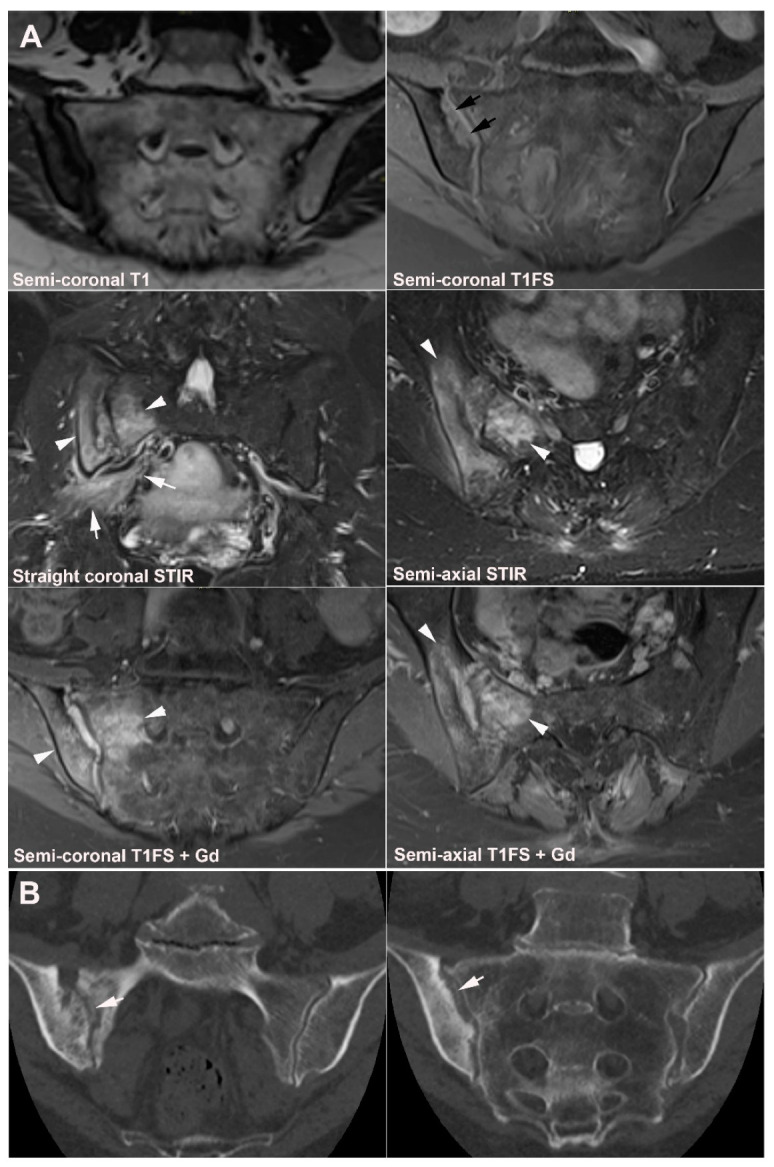
**CPPD changes** in a 72-year-old woman with inflammatory low back and buttock pain for three weeks, starting rather suddenly and with accompanying elevated CRP. (**A**) MRI, semi-coronal T1 and T1FS (upper image row), straight coronal and semi-axial STIR (middle row), and postcontrast semi-coronal and semi-axial T1FS showing pronounced edema and enhancement in the right joint space and the subchondral bone (arrowheads) with concomitant edema in the surrounding soft tissue, especially inferiorly to the joint (white arrows). Irregular iliac joint margins are seen with an iliac area lacking normal bone texture superiorly (black arrows on T1FS). (**B**) Supplementary CT and semi-coronal reconstructions revealed mineralization corresponding to the cartilage (arrows) indicating crystals in addition to an iliac area with subchondral bone resorption mimicking changes seen in hyperparathyroidism, which was not supported biochemically. It was revealed that the patient also had mineralization characteristic of CPPD in the spine and sternoclavicular joints (not shown).

**Figure 28 jcm-12-01039-f028:**
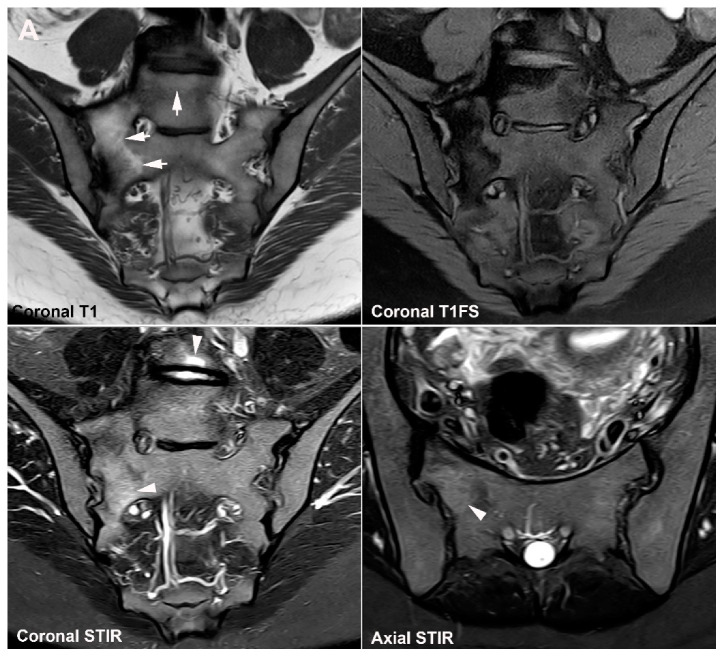
**CNO in adulthood (SAPHO—**Synovitis, Acne, Pustulosis, Hyperostosis), MRI in a 35-year-old woman with intermittent flares of low back pain since the age of 19 years. (**A**) MRI at the age of 35 years during a flare, semi-coronal T1, T1FS and STIR, and semi-axial STIR showing changes confined to the sacral bone in the form of right-sided sacral hyperostosis with manifest fat metaplasia in the bone marrow accompanied by discrete fat metaplasia in the sacrum at the intervertebral (iv) space L5/S1 (arrows), signs of previous osseous inflammation. There is discrete subchondral BME in S2, but the most manifest sign of activity is in the subchondral bone in L5 at the iv space L5/S1 (arrowheads). (**B**) MRI at the age of 19 years, axial T1, and postcontrast T1FS at the level of S1 (upper image row) and upper part of S2, showed manifest inflammation in the sacral bone with enhancement in the adjacent soft tissue (arrows), changes simulating tumor, but biopsy revealed non-specific subacute inflammatory changes and negative cultures.

**Figure 29 jcm-12-01039-f029:**
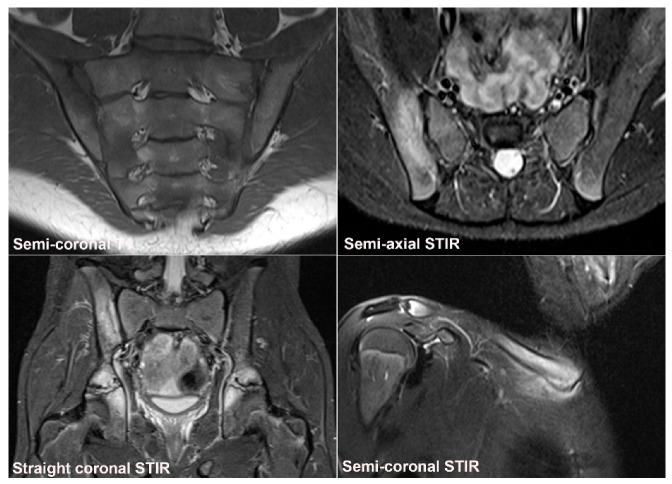
**CNO in childhood** (**CRMO**—chronic recurrent multifocal osteomyelitis). MRI in a 14-year-old boy with recurrent hip and ACW pain for 9 months, semi-coronal T1, and semi-axial STIR of the SIJs, straight coronal STIR of the pelvis, and semi-coronal STIR of the right clavicle showing pronounced BME in the right iliac bone adjacent to the SIJ, but no pathological BME in the sacrum. In addition, pronounced BME around the ischiopubic synchondroses and in the right clavicle.

**Figure 30 jcm-12-01039-f030:**
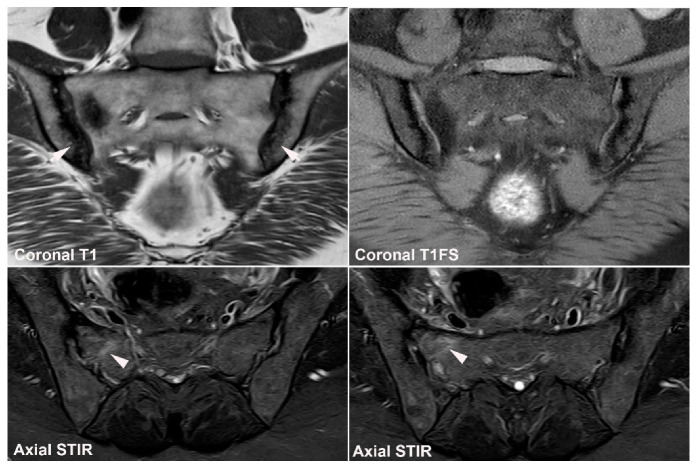
**Hyperparathyroidism** in a 54-year-old man with renal insufficiency complaining of low back pain. MRI, semi-coronal T1, and T1FS and two semi-axial STIR images at S1 (left image) and S2, respectively, showing irregular iliac joint margins with subchondral sclerosis intermingled with small fat depositions (arrows). There are well-delineated sacral joint margins, but an area with BME in the right side of S1 (arrowheads).

**Figure 31 jcm-12-01039-f031:**
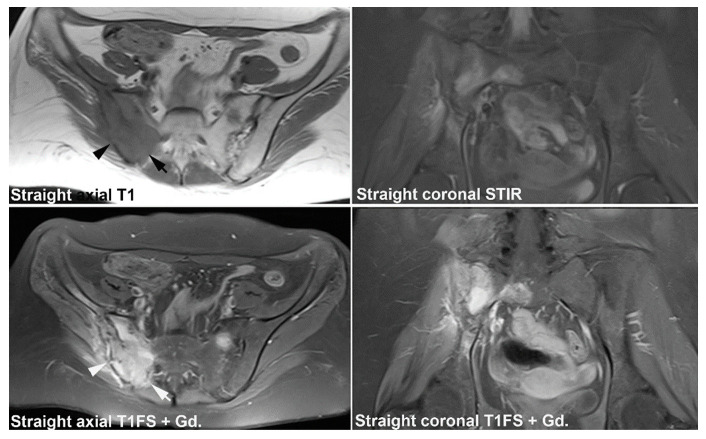
**Malignant tumor**, MRI in a 20-year-old woman, straight axial T1 and coronal STIR (upper image row), and postcontrast T1FS sequences in the same scan planes (lower image row). Osseous destruction of the iliac and sacral bone at the right SIJ is seen with fracture lines in the ileum (arrowheads) and a solid enhancing soft tissue mass posteriorly (arrows). Biopsy confirmed malignancy (osteosarcoma).

**Table 1 jcm-12-01039-t001:** Reported final diagnoses in patients with low back pain and suspected sacroiliitis examined by MRI [17,18].

	Rheumatologist Referral, *n* = 691 *		Referral from Different Specialties, *n* = 281 **		Combined Data, *n* = 972	
Age, mean, years	36		44			
Males/females	261/430		116/165		377/595	
	** *n* **	**%**	** *n* **	**%**	** *n* **	**%**
**SpA**	249	36	71	25	320	33
**Normal SIJ findings**	285	41	123	44	408	42
**Alternative ** **SIJ-related diagnoses**	130	19	87	31	217	22
Anatomic variants	41 ^#^	5.9 ^#^	15 ^##^	5.3 ^##^	56	5.7
OCI	17	2.5	25	8.9	42	4.3
Degenerative SIJ findings	25	3.6	12	4.3	37	3.8
SIJ DISH findings	24	3.5	4	1.5	28	2.9
Septic sacroiliitis/discitis	4	0.6	15	5.3	19	2.0
Stress reaction/fracture	8	1.2	2	0,7	10	1.0
Tumor	11	1.6	1	0.3	12	1.2
**Spine disorders**						
Degenerative spinal disease	305 ^¤^	44.1 ^¤^	11 ^¤¤^	4 ^¤¤^	316	33
Other spinal disorders	10	1.5	2	0.7	12	1.2

* Disease start specified to be before 40 years of age. MRI sequences: coronal T1 and STIR of the SIJ and an axial pelvic STIR sequence. No dedicated spine imaging. ** MRI sequences: coronal T1 and T2FS of the SIJ in addition to coronal T1FS before and after intravenous contrast; lumbar spine: sagittal T1 and T2 sequences. ^#^ Only hemivertebra; ^##^ only hemivertebra and accessory SIJ. ^¤^ Any changes by coronal SIJ imaging and an axial pelvic STIR sequence; ^¤¤^ Substantial spinal changes. N = number; SpA = spondyloarthritis; OCI = osteitis condensans ilii; DISH = diffuse idiopathic skeletal hyperostosis.

## Data Availability

No primary data was generated in this review.

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
