# Peer review of "Diagnostics of Sacroiliac Joint Differentials to Axial Spondyloarthritis Changes by Magnetic Resonance Imaging"

_jcm, 2023, doi:10.3390/jcm12031039_

Round 1
Reviewer 1 Report
The author presents a strong and comprehensive review of MRI findings at the sacroiliac joints, focusing on differential diagnoses of BME and other findings. Especially the mid-section about unfused nuclei is compelling and presents novel points. Thus, we find good teaching material here. Therefore, the article is not only entertaining but also instructive for readers of different experice levels. Please find below a few suggestions:
Major issues:
Ma: Honestly, I would very much like to see at least one case of prototypical axSpA so that the reader is reminded what we are talking about.
Ma: I suggest critically discussing the studies that show an equal distribution of sexes in nr-axSpA. Those studies might overemphasise MR findings and the - not very specific - clinical arm of the ASAS classification criteria. At least stress these results less in the abstract and introduction.
Ma: I recommend referring to the latest proposal of the ASAS group for conducting a standardised protocol. This protocol is as described in the text but uses the term "erosion specific" sequence and implies that a 3D gradient echo sequence is preferred over a 2D T1fs.
Ma: I would also like to emphasise the correct planning of the image stack. While it is true that "the most anterior coronal slice will typically be in middle portion of the cartilaginous joint compartment" this is not a desired outcome of the MRI examination. The image stack should be large enough to cover the whole SIJ. Therefore, the most anterior slice should not show any SIJ whatsoever.
Ma: "Dysmorphic SIJ will usually not cause diagnostic problems because the altered joint form is directly visible (Figure 6F)"... To my knowledge, SIJ dysmorphism is not well-established in the average rheumatology and radiology community. While MSK experts might be aware of this problem, I see misclassified BME in patients with anatomical variations and similar conditions on a weekly basis. Furthermore, it is unclear whether mechanical strain based on anatomical variations can trigger and maintain active inflammation in patients susceptible to SpA - or is simply a mimicker of the disease, as presented in this review. There are at least some mouse models suggesting that mechanical load promotes inflammation in axSpA.
Minor issues:
Mi: Other mechanic changes: The localisation with the "superior/middle anterior joint" description could be rephrased to be more precise and match the excellent accompanying figure.
Mi: In my experience, vacuum phenomenon is abundant in healthy controls, axSpA patients and degenerative conditions. However, I do not have published data to support this.
Mi: Bone resorption in figure 24 is quite pronounced and may not be as typical for CPPD as it is for hyperparathyroidism. However, this is an excellent case!
Author Response
The author presents a strong and comprehensive review of MRI findings at the sacroiliac joints, focusing on differential diagnoses of BME and other findings. Especially the mid-section about unfused nuclei is compelling and presents novel points. Thus, we find good teaching material here. Therefore, the article is not only entertaining but also instructive for readers of different experice levels. Please find below a few suggestions:
Response
Thank you for finding the manuscript interesting.
Major issues:
Ma: Honestly, I would very much like to see at least one case of prototypical axSpA so that the reader is reminded what we are talking about.
Response
I agree, it is advantageous to introduce readers not familiar with imaging of spondyloarthritis to the typical changes. I have therefore included images of two patients with non-radiographic sacroiliitis and predominant active and predominant structural lesions, respectively, but have not included an extensive description of the typical findings with concomitant references as the manuscript and number of references already is extensive.
Ma: I suggest critically discussing the studies that show an equal distribution of sexes in nr-axSpA. Those studies might overemphasise MR findings and the - not very specific - clinical arm of the ASAS classification criteria. At least stress these results less in the abstract and introduction.
Response
The reference to the equal distribution of sexes in nr-axSpA is, as mentioned not based on specific analyses of differences between radiographic and nr-axSpA. I was chosen because it was already used as a reference to the non-radiographic axSpA. However, there are other cohort studies showing similar sex distribution, the most appropriate to be added is the Canadian study by Wallis et al. which therefore has been included in the reference list.
Ma: I recommend referring to the latest proposal of the ASAS group for conducting a standardised protocol. This protocol is as described in the text but uses the term "erosion specific" sequence and implies that a 3D gradient echo sequence is preferred over a 2D T1fs.
Response
The standardised protocol proposed by the ASAS group has not been published as a manuscript, but only as abstracts in connection to the EULAR and ACR congress. I therefore referred to two publications stating the recommendation. The congress abstracts were identical regarding proposed cartilage sequences, T1-weighted with fat suppression (2D or 3D), which is also stated in the manuscript by Diekhoff et al. (reference 34). However, in the publication by Poddubnyy et al. (reference 35) a gradient echo sequence is recommended, mainly based on German studies. The ASAS recommendation of T1FS is probably chosen to secure that the cartilage sequence can be done on all scanners. For clarification the EULAR abstract has been included as a reference although I do not disagree with you that a 3D gradient echo sequence is preferred over a 2D T1FS sequence, but it may require 3T and/or modern MR-scanners. The choice of a gradient echo sequence has been added, but also that the cartilage sequence used depends on the MR-equipment.
Ma: I would also like to emphasise the correct planning of the image stack. While it is true that "the most anterior coronal slice will typically be in middle portion of the cartilaginous joint compartment" this is not a desired outcome of the MRI examination. The image stack should be large enough to cover the whole SIJ. Therefore, the most anterior slice should not show any SIJ whatsoever.
Response
I agree that the coverage should be larger than the joints which has been added in the manuscript.
Ma: "Dysmorphic SIJ will usually not cause diagnostic problems because the altered joint form is directly visible (Figure 6F)"... To my knowledge, SIJ dysmorphism is not well-established in the average rheumatology and radiology community. While MSK experts might be aware of this problem, I see misclassified BME in patients with anatomical variations and similar conditions on a weekly basis. Furthermore, it is unclear whether mechanical strain based on anatomical variations can trigger and maintain active inflammation in patients susceptible to SpA - or is simply a mimicker of the disease, as presented in this review. There are at least some mouse models suggesting that mechanical load promotes inflammation in axSpA.
Response
Thank you for this relevant comment. The wording about dysmorphic changes has been changed accompanied by an illustration of a case where supplementary CT was deemed necessary to exclude erosions.
Minor issues:
Mi: Other mechanic changes: The localisation with the "superior/middle anterior joint" description could be rephrased to be more precise and match the excellent accompanying figure.
Response
Our studies based on two slice orientations have constantly shown that the frequency of mechanically load and strain-related changes is highest in the anterior middle joint portion, but in the study by Renson 2022 based on semi-coronal slices, they found the superior portion to be the most frequent site. The difference is due to the interpretation of the joint location based on one and two slice orientations, respectively. It has now been more clearly stated in the manuscript and “superior/middle” has been omitted..
Mi: In my experience, vacuum phenomenon is abundant in healthy controls, axSpA patients and degenerative conditions. However, I do not have published data to support this.
Response
The occurrence of vacuum phenomenon in the sacroiliac joint is frequent and often accepted as a normal finding, especially in elderly. There is an old study of Peh and Ooi, 1997 (doi: 10.1097/00007632-199709010-00013), analyzing the frequency in a control group with a mean age of 69.2 years (range 55-84 years). VP was found in 41 of 60 control patients (68.3%) and in 71 of 120 SIJ (59.2%). Usually, VP is grouped together with other signs of joint degeneration by CT, also being the case in the manuscript which therefore is unchanged regrading this aspect.
Mi: Bone resorption in figure 24 is quite pronounced and may not be as typical for CPPD as it is for hyperparathyroidism. However, this is an excellent case!
Response
Thank you for liking the case. It has been added that there is an iliac area with subchondral bone resorption mimicking changes seen in hyperparathyroidism, which was not supported biochemically.
I hope the changes performed are satisfactory.

Reviewer 2 Report
The author presents a thematically interesting and excellently written review on important and rare differential diagnoses to axial spondylarthritis in MR imaging. One notices an exceptionally deep understanding of the author about the underlying pathogenesis of the disease, which is also didactically reflected in the thematically meaningful structure of the manuscript, ranging from the detailed description of aspects of development in children and adolescents, normal anatomy and its variants to the presentation of rare differential diagnoses. A large number of very good illustrations are included. Overall, the manuscript reads like a chapter of a dedicated medical textbook.
To further enhance the manuscript's relevance to readers, I suggest the following major additions:
- Even if the presented manuscript is dedicated to the differential diagnoses of axSPA, a comparative chapter on imaging diagnosis of axSPA should be included to directly compare the important aspects in reporting with the differential diagnoses and illustrations presented here. An additional table on expected imaging findings compared to axSPA would also be an enriching feature
- There is a lack of presentation on currently improved or more advanced, e.g., quantitative or semi-quantitative, MRI sequences that could potentially facilitate the differential diagnosis of axSPA.
Author Response
The author presents a thematically interesting and excellently written review on important and rare differential diagnoses to axial spondylarthritis in MR imaging. One notices an exceptionally deep understanding of the author about the underlying pathogenesis of the disease, which is also didactically reflected in the thematically meaningful structure of the manuscript, ranging from the detailed description of aspects of development in children and adolescents, normal anatomy and its variants to the presentation of rare differential diagnoses. A large number of very good illustrations are included. Overall, the manuscript reads like a chapter of a dedicated medical textbook.
Response
Thank you for the positive evaluation
To further enhance the manuscript's relevance to readers, I suggest the following major additions:
- Even if the presented manuscript is dedicated to the differential diagnoses of axSPA, a comparative chapter on imaging diagnosis of axSPA should be included to directly compare the important aspects in reporting with the differential diagnoses and illustrations presented here. An additional table on expected imaging findings compared to axSPA would also be an enriching feature
- There is a lack of presentation on currently improved or more advanced, e.g., quantitative or semi-quantitative, MRI sequences that could potentially facilitate the differential diagnosis of axSPA.
Response
A comparative chapter on diagnostic imaging in axSpA could be advantageous, but would add to the volume of the manuscript and the number of references which currently are at the limit of extend. As it will be really advantageous to introduce readers not familiar with imaging of spondyloarthritis to typical SpA changes, images of two patients with non-radiographic sacroiliitis and predominant active and predominant structural lesions, respectively, have been included. I hope you agree that this change is sufficient.
The potential possibilities of semi-quantitative MRI sequences to facilitate the differential diagnosis of axSpA is included by referring to the cut-off values stated in the data-driven ASAS publication by Maksymowych et al. (doi:10.1093/rheumatology/keab099). To my knowledge, there is currently no publications supporting the potential use of quantitative MR-sequences in the differential diagnosis except the well-known vale of DWI to detect malignancies. Differentiation from malignancies is usually not a major problem. Therefore, only the following has been included in the description of technical aspects: “More advanced MR-sequences, such as DWI (diffusion weighted imaging) has been studied without confirming certain diagnostic advantages compared to the sequences mentioned above”.
I hope the changes performed are satisfactory.

Round 2
Reviewer 2 Report
The changes made to the manuscript during revision are sufficient and the paper appears acceptable for publication in its revised form.